# FORMALMATH: BENCHMARKING FORMAL MATHEMATICAL REASONING OF LARGE LANGUAGE MODELS

## ABSTRACT

Formal mathematical reasoning remains a significant challenge for artificial intelligence, constrained by the limited scope and scale of existing benchmarks. To address this, we introduce FormalMATH, a comprehensive Lean4 benchmark consisting of 5,560 formally verified problems, meticulously curated through human-in-the-loop methods. This benchmark encompasses a broad range, from high-school Olympiad challenges to undergraduate-level theorems across diverse domains, including algebra, applied mathematics, calculus, number theory, and discrete mathematics. Our evaluation of state-of-the-art LLM-based theorem provers reveals notable limitations: even the leading model, DeepSeek-Prover-V2, achieves only a 28.31% success rate under practical sampling budgets, displaying marked domain bias (e.g., excelling in algebra while struggling with calculus) and an over-reliance on simplified automation tactics. We find that test-time scaling (e.g., Pass@1024) offers only marginal performance gains on FormalMATH, pointing to a critical lack of exploration during training. Additionally, we observe a counterintuitive inverse relationship between natural-language solution guidance and proof success in chain-of-thought reasoning scenarios, indicating that human-written informal reasoning introduces noise rather than clarity in formal contexts. Analysis of common error patterns among existing provers highlights issues such as the misuse of automatic tactics (*e.g.*, `aesop`), difficulties in managing complex inequalities, and redundant hypothesis introduction. We assert that FormalMATH provides a robust platform for benchmarking formal mathematical reasoning capabilities.

## 1 INTRODUCTION

Formal mathematical reasoning (FMR) (Yang et al., 2024) represents a specialized form of mathematical practice grounded in formal systems (Leino, 2010; Mathlib Community, 2020; Barras et al., 1997), which provides a rigorous axiomatic framework essential for automated proof validation. However, FMR is inherently challenging for humans. For instance, the Liquid Tensor Experiment (Scholze, 2022) and the Polynomial Freiman-Ruzsa Conjecture (Tao, 2023) have taken years of effort by human experts to formalize and yet remain incomplete. Recent works have leveraged self-supervised learning (Polu & Sutskever, 2020), chain-of-thought (CoT) finetuning (Xin et al., 2024), and scalable tree-search (Xin et al., 2025) to explore complex proof strategies, demonstrating the significant potential of large language models (LLMs) for FMR. While there are several formal mathematics benchmarks, such as MiniF2F (Zheng et al., 2021) and ProofNet (Azerbayev et al., 2023) that are widely used to evaluate the FMR capabilities of LLMs, they still present a few critical limitations: (1) Scope limitation: Existing benchmarks are narrowly scoped. For instance, MiniF2F is restricted to high school-level algebra and number theory, while ProofNet focuses narrowly on undergraduate-level analysis and algebra. Their narrow scopes limit the capacity to evaluate holistic FMR capabilities across diverse mathematical domains. (2) Dataset size: Formal mathematics benchmarks remain relatively small in scale. MiniF2F contains merely 244 problems in its test set, and ProofNet includes only 186. This constrains benchmarking robustness and hinders the development of generalizable FMR systems. (3) Performance Saturation: State-of-the-art theorem provers, such as Kimina-Prover (Wang et al., 2025), now achieve success rates exceeding 80.7%, signaling that existing benchmarks may be nearing their practical utility limits.

To address these limitations, we introduce FormalMATH, a large-scale Lean4 (Moura & Ullrich, 2021)-based benchmark containing 5,560 formally verified mathematical statements. FormalMATH

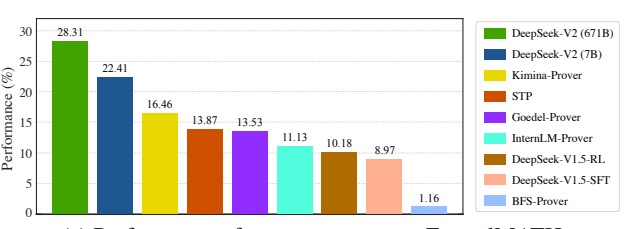 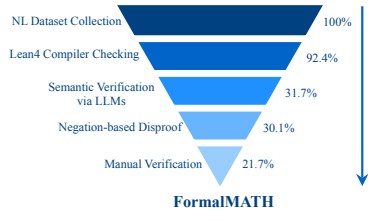

(a) Performance of current provers on FormalMATH    (b) Data preservation rate

Figure 1: (a) Performance comparison of existing theorem provers on the full FormalMATH benchmark. Results show Pass@1×32×100 accuracy for best-first-search-based (BFS) methods, including BFS-Prover and InternLM-Prover, and Pass@32 accuracy via single-pass generations (SPG) for the other provers, including Kinima-Prover, STP, Goedel-Prover, DeepSeek-V1.5-RL, and DeepSeek-V1.5-SFT. (b) Funnel chart illustrating the percentage of data that is preserved after each filtering stage in our human-in-the-loop pipeline.

includes a broad spectrum of mathematical domains, such as algebra, geometry, calculus, number theory, discrete mathematics, and more, while simultaneously spanning multiple difficulty levels, ranging from high school olympiad problems to undergraduate-level theorems.

We evaluate state-of-the-art LLM-based theorem provers on the FormalMATH benchmark, revealing significant challenges for these systems. For instance, the best-performing model — Kimina-Prover (Wang et al., 2025) achieves only 16.46% on FormalMATH under the pass@32 metric, while BFS-Prover (Xin et al., 2025) attains just 11.13% using a best-first search with a sampling budget of $1 \times 32 \times 100$. Our analysis of these results yields several intriguing insights. First, existing provers exhibit a pronounced domain bias, excelling primarily in high-school-level algebra and applied mathematics while struggling with other mathematical domains. This highlights critical gaps in their cross-domain generalizability. Second, the provers frequently reduce multi-step reasoning to single-tactic invocations (*e.g.*, "aesop" (Limperg & From, 2023) and "linearith"), bypassing necessary deductive rigor. Third, while CoT reasoning (Wei et al., 2022) enhances performance on FormalMATH statements, adding natural language solutions reduces success rates, suggesting such guidance introduces ambiguity rather than clarity. Our contributions include:

- **A Large and Comprehensive Lean4 Benchmark**: We present FormalMATH, a benchmark comprising 5,560 formally verified mathematical statements covering diverse subdomains, including high-school olympiad and college-level problems. The dataset is dual-reviewed by 12 human experts with multiple large language models in the loop to ensure correctness. FormalMATH is $22.8\times$ larger than the widely used MiniF2F benchmark.

- **Comprehensive Evaluation of LLM-based Theorem Provers**: Our systematic evaluation highlights key limitations in state-of-the-art theorem provers: 1. Even the best-performing model achieves only a 28.31% success rate on FormalMATH, 2. Existing provers show significant domain bias, excelling in areas like algebra but underperforming in others, such as calculus, 3. A counterintuitive inverse relationship emerges where providing natural language solution guidance reduces proof success rates in chain-of-thought scenarios.

- **Identification of Common Error Patterns in Lean4 Proving** 1. Provers often resort to inappropriate automatic tactics when facing unsolvable problems. 2. Provers frequently produce incomplete proofs with meaningless placeholders to simplify tail behavior, rather than rigorously addressing them. 3. Provers struggle to solve complex inequalities using tools like nlinarith. 4. Provers tend to generate redundant hypotheses. These limitations highlight key areas for enhancing LLM-based provers.

## 2 RELATED WORK

**Formal Mathematical Reasoning.** (Xin et al., 2024; Lin et al., 2025; Dong & Ma, 2025) utilize LLMs to generate entire proofs directly. These methods then typically employ techniques like best-of-N sampling to scale up test-time computation, often achieving results comparable to proof-search methods. As a SPG method, Kimina-prover (Wang et al., 2025) employs long-CoT (Guo et al., 2025) with a think prompt template during reinforcement learning (Team et al., 2025), achieving impressive performance. Section 5.1 compares various sampling approaches on FormalMATH.

| Domain | Undergraduate Domains | | | | | | |
|---|---|---|---|---|---|---|---|
| | Int | Pre | Alg | Diff | MC | SS | Other |
| **Percentage (%)** | 30.8 | 18.9 | 15.9 | 11.0 | 9.6 | 8.9 | 4.9 |
| Domain | High School Domains | | | | | | |
| | Alg | NT | AM | DM | Geo | Pre | Cal | Other |
| **Percentage (%)** | 50.2 | 22.3 | 9.0 | 8.6 | 5.8 | 1.9 | 1.9 | 0.3 |

Table 2: Domain distribution in FormalMATH-Lite dataset. Abbreviations: Int=Integral, Pre=Precalculus, Alg=Algebra, Diff=Differentiation, MC=Multivariable calculus, SS=Sequences Series, NT=Number Theory, AM=Applied Mathematics, DM=Discrete Mathematics, Geo=Geometry, Cal=Calculus.

**Formal Theorem Proving Benchmarks.** Benchmarks for assessing Lean4-based theorem-proving capabilities can be categorized based on whether they use off-the-shelf formal proofs. Benchmarks derived from existing libraries, such as LeanDojo (Yang et al., 2023), extract proofs and theorems from the off-the-shelf Lean Mathlib library (Mathlib Community, 2020). Lean-workbook (Ying et al., 2024a) is another large dataset in Lean4. It is used as a training dataset—for example,

| Benchmark | # Problems | Difficulty |
|---|---|---|
| MiniF2F | 244 | Olympiad |
| ProofNet | 186 | Undergraduate (UG) |
| FIMO | 149 | Olympiad |
| PutnamBench | 522 | Olympiad |
| ProverBench | 325 | Olympiad |
| FormalMATH | **5,560** | **Olympiad & UG** |

Table 1: Comparison of existing Lean4 benchmarks.

provers like InternLM-Prover (Ying et al., 2024b) use expert iteration to train on it. In contrast, benchmarks without pre-formalized proofs operate under a different paradigm. Instead of providing reference proofs, these benchmarks present only formalized problem statements, often derived from informal mathematics. Proving systems are used to generate a proof from scratch, the validity of which is then verified using the Lean compiler (Leanprover Community, 2023). As shown in Table 1, representative benchmarks include: (1) MiniF2F (Zheng et al., 2021), which compiles 244 competition-level problems from AMC, AIME, and IMO in its test dataset, (2) ProofNet (Azerbayev et al., 2023), which comprises 186 problems from undergraduate-level analysis and algebra, (3) FIMO (Liu et al., 2023), which contains 149 IMO shortlist problems, and (4) Putnam-Bench (Tsoukalas et al., 2024), which is a benchmark of 522 Lean4 problems from the Putnam competition. FormalMATH falls into the category of Lean4 dataset that require new proof completion, comprising 5,560 diverse problems formalized from high-school competition-level sources (*e.g.*, Omni-Math (Gao et al., 2024) and BlueMO (Zhang et al., 2024)) and undergraduate-level problems (*e.g.*, U-Math (Chernyshev et al., 2024), Hardmath (Fan et al., 2024), and DEMIMATH (Demidovich, 1964)).

## 3 FORMALMATH: A LARGE FORMAL MATHEMATICAL REASONING BENCHMARK

### 3.1 OVERALL DATASET STATISTICS

FormalMATH is a rigorously validated Lean4 benchmark comprising 5,560 mathematical statements, each independently verified through a hybrid pipeline combining multi-LLM semantic verification and careful review by Olympiad-level human experts. It is inherently difficult; for example, the state-of-the-art prover (*e.g.*, DeepSeek-V2 (671B)) achieves only a 28.31% Pass@32 score on the benchmark (see Figure 1). The problems span a broad difficulty spectrum, ranging from high-school competition questions in disciplines such as algebra, number theory, discrete mathematics, and geometry, to undergraduate-level challenges in specialized areas including calculus (integration and differentiation), linear and abstract algebra, and sequences and series (see Table 2).

### 3.2 DATA FORMAT

FormalMATH provides a Lean4-compatible JSON format dataset (See Table 3). It supports various tasks in formal mathematical reasoning, including autoformalization, proof completions, etc.

| Field | Type | Description |
|---|---|---|
| **source** | string | Origin of the problem (e.g., Omni-Math, U-Math, Hardmath, etc.) |
| **informal_statement** | string | Human-readable natural-language statement of the problem |
| **formal_statement** | string | The formal Lean 4 statement |
| **informal_proof** | string | Reference informal proof in natural language |
| **domain** | string | Mathematical domain (e.g., algebra, number theory, calculus, etc.) |
| **difficulty** | float | Difficulty score assigned by LLM following the rubric of Gao et al. (2024). |

Table 3: Format of the FormalMATH dataset.

## 3.3 DATA COLLECTION PROCESS

We initially collected informal mathematical statements from a variety of existing high-quality sources (*e.g.*, Omni-Math, BlueMO, HardMath, etc.). To prevent overlap with prior formalization benchmarks, we performed rigorous data decontamination, removing any statements that already appeared in datasets such as LeanWorkbook (Ying et al., 2024a) or MiniF2F (Zheng et al., 2021). The cleaned informal statements are then undergo autoformalization into Lean4 using LLMs, followed by extensive manual quality assurance through a human-in-the-loop process (detailed in Section 4).

## 4 THE PROPOSED HUMAN-IN-THE-LOOP PIPELINE FOR DATA COLLECTION AND FILTERING

The overall pipeline to autoformalize the dataset used in FormalMATH is exhibited in Figure 2.

**Supervised Fine-tuning.** During the development of FormalMATH, we find that mature, open-source autoformalization tools are scarce. To fill this gap, we build our own pipeline on top of two types of LLMs: coding-specialized LLMs (*e.g.*, Qwen2.5-7B-Coder (Bai et al., 2023)) and pre-trained theorem-proving LLMs (*e.g.*, Deepseek-prover-base (Xin et al., 2024)). We then generate training data by having a general-purpose LLM (*e.g.*, GPT-4 (OpenAI, 2023)) iteratively translate natural-language statements into Lean4 statements. Each candidate statement is then passed to the Lean4 compiler, and only those that are type-checked will be kept. This straightforward "compile-and-filter" strategy yields a high-quality corpus of 9,260 paired training examples, which is eventually used to finetune our own autoformalization models.

| Item | Value |
|---|---|
| # Annotators | 12 |
| Preservation rate | 72.09% |
| Cost/statement | $6.89 |
| Total duration | 22 days |

Table 4: Annotation statistics.

**Autoformalization.** For each of the $K$ autoformalizers (implemented by LLMs), we employ a best-of-N sampling strategy (Wang et al., 2022) to generate $N$ formal candidate statements $\mathbf{T}_n^{(k)}$, where $k \in \{1, \ldots, K\}$ denotes the autoformalizer index, and $n \in \{1, \ldots, N\}$ represents the candidate statement index of the $k$-th autoformalizer. All candidate statements $\mathbf{T}_n^{(k)}$ are first validated for syntactic correctness using the Lean4 compiler. Only syntactically valid statements are preserved for subsequent semantic verification.

**Semantic Verification via LLMs.** We implement a semantic verification strategy based on multiple powerful general-purpose LLMs (*e.g.*, o1-mini (Jaech et al., 2024), claude-3.5-Sonnet) to evaluate semantic alignment between natural language mathematics problems and their Lean4 formalizations. Each model employs chain-of-thought reasoning (See the prompt in Appendix G) to complete the following procedures: (1) back-translate Lean4 statements into natural language, (2) compare reconstructed descriptions with original problems, and (3) provide binary judgments (*i.e.*, aligned/misaligned). Importantly, only Lean4 statements that passed semantic verification performed by all the LLMs would be collected. This strategy is guided by the insight that translating Lean4 statements to natural language is a much easier task than the reverse process, and general-purpose LLMs excel at understanding natural language phrasings (Wu et al., 2022). Overall, this procedure filters out 60.7% of syntactically correct but semantically misaligned statements (*i.e.*, from 92.4% to 32.7%). Interestingly, we find distinct consensus patterns across problem difficulty levels – around 30% unanimous agreement rate for high school competition problems and significantly lower consensus for undergraduate-level formalizations (*e.g.*, 4.63% on HardMath).

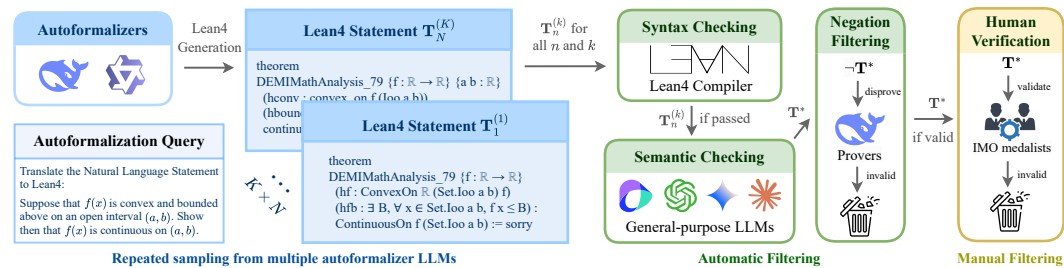

Figure 2: A human-in-the-loop pipeline for formal mathematical statement creation and filtering.

**Disproving a Statement by Proving Its Negation.** Inspired by the Law of the Excluded Middle (LEM (contributors, 2025)), we further filter out certain non-provable formalizations using off-the-shelf LLM-based provers (*e.g.*, DeepSeek-Prover-V1.5). For any formalized statement $\mathbf{T}_n^{(k)}$, we perform the following steps: (1) construct logical negation: construct its logical negation by applying transformation rules such as De Morgan dualization to generate $\neg\mathbf{T}_n^{(k)}$, and (2) automated proof attempts: perform automated proof attempts on $\neg\mathbf{T}_n^{(k)}$ within the formal system $\mathcal{S}$ (*i.e.*, Lean4 compiler). A successful proof of $\neg\mathbf{T}_n^{(k)}$ implies that the original statement $\mathbf{T}_n^{(k)}$ cannot hold on $\mathcal{S}$. Appendix N illustrates the Lean 4 formalization of a number-theoretic conjecture and its negation. By constructing the negation of a statement and applying an LLM-based prover for disproof, the system identifies inconsistencies through boundary case testing (*e.g.*, n = 7) and derives contradictions via systematic case analysis (*i.e.*, `interval_cases`). This strategy has filtered out a few unprovable statements, accounting for 1.6% of the total statements.

**Expert Verification.** We recruited 12 International Mathematical Olympiad medalist-level human experts to manually verify the semantic alignment between natural language statements and their Lean4 formalizations. To ensure reliable human validation, we developed a checklist targeting common error patterns in Lean4 statements and employed the previously introduced multi-LLM-as-judge method to assist human validators. We used cross-validation among humans to further ensure consistency. Table 4 presents key metrics from the human validation stage. Our results demonstrate that the multi-LLM autoformalization and validation pipeline is highly effective, retaining 72.1% of statements from the final stage of LLM-based semantic verification (reducing the error rate from 30.1% to 21.7%) while significantly reducing manual verification efforts. Ultimately, we successfully formalized 21.7% of syntactically and semantically correct mathematical statements from a diverse collection of mathematical problems sourced from multiple data sources. See Appendix B and D for further details.

| Error Category | Description |
| --- | --- |
| **Definition** | Failure to semantically map mathematical entities one-to-one with the refined statement, e.g., missing domain-specific constraints (geometry: points, lines, angles), mismatched variable types, or definitions lacking mathematical significance. |
| **Expressions** | Incorrect arithmetic/logical operations, improper variable use, or misapplication of mathematical/logical rules, e.g., misplaced quantifiers altering the logical structure of the statement. |
| **Constraint Condition** | Constraints that mismatch the problem's requirements, are omitted, or include redundant conditions not specified in the original problem. |
| **Proof Goals** | Proof goals misaligned with the original problem, overly simplified, too general, or incomplete, e.g., omitting parts of the problem like characterizing solutions achieving a maximum value. |

Table 5: Error Pattern Checklist for Autoformalization

# 5 EXPERIMENTS AND DISCUSSIONS

## 5.1 EVALUATING FORMAL THEOREM PROVERS ON FORMALMATH

**LLM-based Prover Settings.** We focus on the following proof-generation approaches:

- **Best-First Tree-Search (BFS) Methods.** Each node in the search tree represents an intermediate proof state, and a heuristic scoring function assigns a priority to each node. We evaluate three baseline models under this category: BFS-Prover (Xin et al., 2025), DeepSeek-Prover-V1.5-RL (Xin et al., 2024), and InternLM-V2.5-Prover (Wu et al., 2024).

| Method | Sampling Budget | | | | |
|---|---|---|---|---|---|
| | 1 | 4 | 8 | 16 | 32 |
| *Best-First Tree Search Methods* | | | | | |
| BFS(InternLM-V2.5) | 7.87 | 15.79 | 20.02 | 22.74 | 25.65 |
| BFS(DS-Prover-V1.5-RL) | 6.81 | 24.70 | 28.47 | 30.65 | 32.71 |
| BFS(BFS-Prover) | 27.10 | 34.04 | 37.56 | 41.75 | 45.88 |
| | **Sampling Budget** | | | | |
| | 32 | 512 | 1024 | 2048 | 3200 |
| *Single-Pass Generation Methods* | | | | | |
| Kimina-Prover-7B | 48.94 | – | – | – | – |
| STP | 48.59 | 51.45 | 52.03 | 52.60 | 53.17 |
| DS-Prover-V1.5-SFT | 40.40 | 44.17 | 45.08 | 46.12 | 46.82 |
| DS-Prover-V1.5-RL | 47.98 | 49.27 | 49.68 | 50.08 | 50.35 |
| Goedel-Prover | 46.70 | 48.68 | 49.04 | 49.20 | 49.41 |
| DS-Prover-V2(7B) | 51.76 | 54.11 | 54.11 | 54.82 | 55.06 |
| DS-Prover-V2(671B) | 56.00 | 60.00 | 61.18 | 61.88 | 61.88 |
| Ensemble of SPG Methods | 54.11 | – | – | – | – |

Table 6: Performance comparison of theorem prover LLMs on FormalMATH-Lite. For BFS methods, the sampling budget values should be multiplied by $N \times 32 \times 100$, where N denotes the number of best-first search attempts.

- **Single-Pass Generation Methods.** The models under this category generate a complete proof in one pass, without iterative refinement or explicit intermediate states. In our paper, we consider the following baseline models: STP (Dong & Ma, 2025), DeepSeek-Prover-V1.5-SFT (Xin et al., 2024), DeepSeek-Prover-V1.5-RL (Xin et al., 2024), Goedel-Prover (Lin et al., 2025), and Kimina-Prover-7B (Wang et al., 2025).

**Metrics: Pass@K.** Pass@$K$ measures the fraction of problems for which at least one valid proof is found among the top $K$ generated attempts. (1) For BFS, $K = N \times S \times T$, where $N$ denotes the number of best-first search attempts, $S$ is the number of tactics proposed during each expansion, and $T$ is the total number of expansion iterations. (2) For SPG, $K$ corresponds to the total number of complete proof trajectories sampled from the model. For each problem $i$, we generate $n_i$ samples and observe $s_i$ successful proofs. The probability that at least one of $K$ randomly selected samples is successful is given by $1 - \binom{n_i-s_i}{K}/\binom{n_i}{K}$, where $\binom{n}{k}$ denotes the binomial coefficient. Pass@$K$ is then computed as:

$$\text{Pass@}K = \frac{1}{n} \sum_{i=1}^{n} \left( 1 - \frac{\binom{n_i-s_i}{K}}{\binom{n_i}{K}} \right), \tag{1}$$

where $n$ is the total number of problems. When $n_i < K$ or $s_i = 0$, the term is set to $0$.

**Prompts.** In the experiments, we only consider vanilla generation strategies (see Example I.1), where models directly generate Lean4 proof without explicit requirement of chain-of-thought (CoT) rationales (natural language thoughts interleaved with Lean4) or augmenting with natural language solutions.

**Finding 1: Existing LLM-based Provers Are Still Far from Solving FormalMATH.** Current LLM-based theorem provers demonstrate unsatisfactory performance on the FormalMATH benchmark under modest sampling budgets. Specifically, one of the current strongest SPG methods, Kimina-Prover, achieves a mere 16.46% under Pass@32, while the best BFS method, BFS-Prover, attains only 11.13% Pass@1$\times$ 32 $\times$100, demonstrating the underlying difficulties of FormalMATH. Notably, both methods use Qwen2.5-Math-7B as their base model but the performance differs dramatically: the former distills curated long-CoT proof traces from a larger LLM-based oracle, and the latter relies on expert iteration via BFS to iteratively enhance the LLM's Lean4 proving abilities.

| Model | Alg | NT | DM | AM | Geo | Pre | Cal | Other |
|---|---|---|---|---|---|---|---|---|
| *High-School Domains* | | | | | | | | |
| DeepSeek-Prover-V1.5-SFT | 11.12 | 5.02 | 4.09 | 14.22 | 5.46 | 9.18 | 1.04 | 0.00 |
| DeepSeek-Prover-V1.5-RL | 12.65 | 5.99 | 4.77 | 14.66 | 7.17 | 10.20 | 3.12 | 0.00 |
| Goedel-Prover | 17.47 | 7.04 | 5.68 | 16.19 | 9.56 | 13.27 | 5.21 | 6.25 |
| STP_Lean | 18.49 | 6.95 | 5.68 | 15.75 | 10.58 | 13.27 | 6.25 | 0.00 |
| Kimina-Prover-7B | 18.64 | 12.15 | 8.86 | 26.04 | 10.92 | 17.35 | 8.33 | 6.25 |
| BFS-Prover | 11.24 | 10.48 | 9.77 | 18.38 | 10.24 | 4.08 | 10.42 | 12.50 |

| | Alg | Pre | Diff | MC | Int | SS | Other |
|---|---|---|---|---|---|---|---|
| *Undergraduate Domains* | | | | | | | |
| DeepSeek-Prover-V1.5-SFT | 38.67 | 17.98 | 1.92 | 7.14 | 0.00 | 0.00 | 0.00 |
| DeepSeek-Prover-V1.5-RL | 41.33 | 19.10 | 1.92 | 9.52 | 0.00 | 0.00 | 0.00 |
| Goedel-Prover | 54.67 | 33.71 | 1.92 | 14.29 | 0.00 | 4.35 | 2.22 |
| STP_Lean | 49.33 | 31.46 | 1.92 | 11.90 | 0.00 | 4.35 | 2.22 |
| Kimina-Prover-7B | 58.67 | 37.08 | 1.92 | 14.29 | 0.00 | 0.00 | 0.00 |
| BFS-Prover | 24.00 | 10.11 | 0.00 | 0.00 | 6.21 | 4.35 | 4.44 |

Table 7: Performance comparison across different mathematical domains on FormalMATH-Lite. Note: Abbreviations: Alg=Algebra, NT=Number Theory, DM=Discrete Mathematics, AM=Applied Mathematics, Geo=Geometry, Pre=Precalculus, Cal=Calculus, Diff=Differentiation, MC=Multivariable Calculus, Int=Integral, SS=Sequence Series.

Methods built upon DeepSeek-Prover-V1.5 exhibit a performance hierarchy that underscores the fundamental limitations of common post-training strategies nowadays. While the DeepSeek-V1.5-SFT baseline achieves 8.97% accuracy, its reinforcement learning (RL) variant improves only marginally to 10.18%, a mere +1.21% gain that exposes the diminishing returns of rule-based sparse reward shaping in complex theorem spaces. However, another more sophisticated training paradigm, STP's self-play curriculum learning, achieves 13.87% (+4.89% over SFT) while Goedel-Prover's expert iteration reaches 13.53% (+4.55% over SFT). Overall, we hypothesize these low success rates on FormalMATH suggest potential limitations of LLM-based provers: (1) reward sparseness: relying solely on binary rewards generalizes complex problems difficult, and techniques like intrinsic rewards may better guide exploration and skill acquisition. (2) combinatorial search complexity: brute-force search and dependency on limited successful reasoning traces to RL and expert iteration affects sample efficiency and effective exploration.

**Finding 2: Provers' Unbalanced Performance Across Mathematical Domains of Formal-MATH.** Table 7 reveals significant domain bias in existing theorem provers. Under Pass@32, Godel-Prover achieves strong performance in algebra-related domains (*e.g.*, 17.47% in high school algebra and 50% in undergraduate algebra) but performs poorly in calculus (5.21%) and discrete mathematics (0%). This imbalance persists at the undergraduate level, with success rates in precalculus (33.71%) far exceeding those in differentiation (1.92%) and integration (0%). We attribute this bias to the training data distributions. Using FormalMATH's domain categorization prompt (see Appendix H), we analyzed Godel-Prover's training corpus by sampling 200 problems. As shown in Figure 6a, the dataset disproportionately emphasizes applied mathematics and algebra (68% combined), while discrete math, number theory, and precalculus collectively constitute less than 5%.

## 5.2 Evaluating Test-time Scaling of Formal Theorem Provers on FormalMATH-Lite

**Experimental Setups.** This section examines test-time scaling of formal mathematical reasoning capabilities in LLM-based theorem provers using a simpler subset of FormalMATH benchmark. For simplicity, we evaluate only BFS and SPG. For each LLM, we use a maximum context length of 32K for reasoning models (e.g., DeepSeek-Prover-V2) and 4096 for other non-reasoning models. The sampling temperature is set to 0.7.

**FormalMATH-Lite.** To enable scalable yet rigorous evaluation, we introduce FormalMATH-Lite, a deliberately simplified and compact subset containing only 425 problems (359 high-school level and 66 undergraduate level). Compared to the full FormalMATH-All dataset, FormalMATH-Lite is intentionally easier and lighter by carefully controlling the proportion of difficult problems and maintaining the original domain distribution. This simpler in difficulties design significantly amplifies the effect of test-time scaling when using large sampling budgets (e.g., Pass@3200).

**Experimental Settings.** In this experiment, we maintain identical experimental configurations to those in Section 5.1, with one critical exception: the sampling budget scales. Section 5.1 used constrained sampling budgets (*e.g.*, Pass@32) due to computational resource limitations of the full FormalMATH benchmark. Here, leveraging FormalMATH-Lite, we deploy expanded sampling budgets (*e.g.*, up to Pass@3200 for SPG and Pass@32×32×100 for BFS).

**Finding 3: Subtle Performance Enhancement via Test-time Scaling.** Table 6 reveals limited returns when applying test-time scaling to formal theorem proving on FormalMATH. For instance, STP achieves only a 4.58% absolute improvement (from 48.59% at Pass@32 to 53.17% at Pass@3200) despite a $100\times$ sampling budget increase. While BFS-Prover demonstrates better scaling dynamics, attaining an 18.78% gain (27.10% via Pass@1×32×100 to 45.88% via Pass@32×32×100), under a $32\times$ budget expansion, however, it still underperforms SPG methods.

Ensembling SPG methods (*i.e.*, via composing STP, Goedel-Prover, DeepSeek-V1.5-SFT, and DeepSeek-V1.5-RL) yields only marginal gains, from 53.17% by STP alone to 54.11% – a mere 0.84% uplift. This is in sharp contrast to the near-linear scaling performance increments in informal reasoning (Muennighoff et al., 2025). In informal mathematics, pseudo-continuous reward signals during sampling create pathways where imperfect reasoning chains, despite their logical flaws, can occasionally "stumble" into correct answers. This suggests that valid conclusions may emerge even when the intermediate steps aren't rigorously sound.

Formal theorem proving lacks such tolerance. A single misplaced tactic or type error invalidates the entire proof trajectory, rendering incremental sampling ineffective. While verifier-guided proof search (*e.g.*, BFS with access to intermediate proof states) theoretically mitigates this brittleness better than SPG methods, current implementations remain computationally impractical and lack scaling efficiency.

## 5.3 CoT Can Enhance Model Capabilities on Formal Mathematical Reasoning

**Finding 4: Naive CoT Outperforms Natural Language Guidance in Formal Theorem Proving.** Across both SFT and RL configurations, we observe a consistent ranking of decoding strategies. Generally, naive CoT attains the highest Pass@K (from K equals 32 to 3200) accuracy, while NL-augmented CoT performs an intermediate position better than vanilla decoding. For example, under $K = 3200$, DeepSeek-V1.5-SFT achieves 50.6% with CoT and 49.2% with NL-augmented CoT and 47.0% with vanilla decoding, and DeepSeek-V1.5-RL achieves 51.7%, 51.2%, and 49.8%, respectively. On the other hand, it appears to be counterintuitive that NL-augmented CoT does not yield superior results compared to simple CoT. Figure 6b reveals a counterintuitive trend in perplexity distributions across prompting strategies: NL-augmented CoT consistently increases model uncertainty compared to naive CoT (*i.e.*, mean perplexity from 1.93 to 5.07) across Lean4 problems.

In Appendix O.3, the failed NL-augmented CoT proof reveals a fundamental error pattern: although the NL outline and the Lean4 script target the same semantic goal, the high-level sketch omits essential parameters and case distinctions that Lean's tactics require. We hypothesize that this discrepancy stems from an intrinsic misalignment between the action space of informal, NL reasoning and the tactic space of Lean4 formalization.

In this particular instance, the NL-augmented CoT followed the NL solution by working on `modulo` `7`, and asserting informally that $x^3 \ mod \ 7 \in \{0, 1, 6\}$ and $y^4 \ mod \ 7 \ \in \{0, 1, 2\}$ but does not materializes those assertions into the fifteen concrete `have _ = const` hypotheses branch that Lean4's decision procedures demand. As a result, when the script invokes tactics (*i.e.*, `omega`) reports that the context simply lacks the linear congruences needed to derive a contradiction.

In contrast, naive CoT autonomously selects the larger prime modulus 13 without human-written prior, interleaves each residue-case split with explicit tactic calls producing hypotheses like `have h` `: ` $x^3$ `% 13 = 5 := by simp [hxy_mod]`, and then immediately discharges each branch with `omega`. By reconstructing its own detailed, tactic-level proof, CoT aligns semantic insight with low-level proof obligations, guaranteeing that every subgoal carries the precise numeric constraints required for full automation–whereas the NL-augmented approach, despite being semantically correct at a high level, leaves critical tactical steps unstated and thus fails to complete the proof.

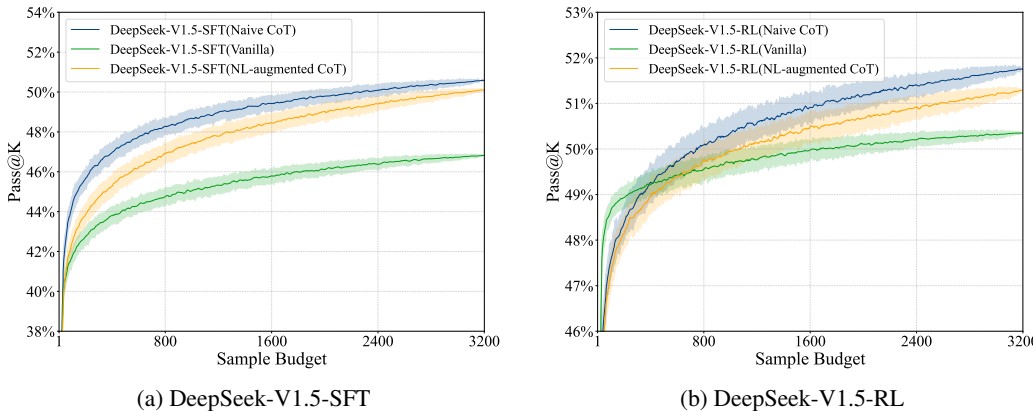

(a) DeepSeek-V1.5-SFT                    (b) DeepSeek-V1.5-RL

Figure 3: Pass@K accuracy curves for DeepSeek-V1.5 provers across different reasoning configurations.

# 6  DELVING INTO COMMON ERROR PATTERNS OF EXISTING PROVERS

## 6.1  ERROR PATTERNS ANALYSIS AND CASE STUDY

We use o4-mini to automate error diagnosis and classification through a two-step process: first extracting key verbalized features (*i.e.*, Example J), then assigning error categories via a second prompt-driven call (*i.e.*, Example K). For each prover, we randomly sampled 100 failed proofs and processed them through this pipeline. Human domain experts then manually reviewed and corrected both the extracted features and preliminary labels. We identified the four most common failure patterns—incomplete proofs, inability to handle complex inequalities, improper use of automation tactics, and redundant hypothesis introduction, as summarized in Table 8. Note that a single proof attempt may exhibit multiple errors, so percentages do not sum to 100%.

**Improper Use of Automation Tactics.** Existing LLM-based Lean4 provers frequently generate proofs that rely heavily on automation tactics – such as `aesop` (Limperg & From, 2023), `simp`, and `linarith`, to streamline the low-level, step-by-step reasoning required by tactic-based proofs. For example, `aesop` performs a best-first proof search over a database of tagged lemmas and applies rewriting, splitting, and instance search to discharge goals. But these tactics depend on fixed heuristics and pre-tagged lemmas that may not match the structure of every proof: when over-invoked or misconfigured, they can dramatically expand the search space, lead to nontermination or timeouts, or even transform goals into irrelevant or unsolvable forms. In particular, automated tactics often struggle to supply the explicit constructions or witnesses required by truly constructive proofs (Smith, 1995), which may discharge the main proposition without building the underlying data, resulting in incomplete or invalid reasoning. Taking the failed proof of `omni_theorem_4000` (Table 11) as an example, it fails to construct a witness $a$ within the correct domain that satisfies both (1) $a \leq 1 \vee a > 0$ and (2) $f(x) = \begin{cases} 0, & \text{if } x \neq -a^2 \\ a, & \text{if } x = -a^2 \end{cases}$. Instead of performing case-by-case analysis, the proof, however, introduces the incorrect witness $a = 0$, and relies on `simp` to close off the remaining goals that are not designed to solve, without specifically analyzing the core function $(x + y^2) \cdot f(y \cdot f(x)) = x \cdot y \cdot f(y^2 + f(x))$.

**Inabilities to Handle Complex Inequalities.** Current provers over-rely on `linarith` and `nlinarith` to find contradictions between hypotheses that are linear and some non-linear (in)equalities. Common procedures using them require the provers to (1) mix high-degree polynomials and rational functions, (2) exploit cyclic or symmetric structure, and (3) use domain-specific lemmas (*e.g.*, rearrangements, Chebyshev, AM-GM variants). For the failed proof `algebra_528739` (Table 11), `nlinarith` must first clear denominators in the sum of fractions by introducing the common denominator: $D = \left(a^3 + b^3 + abc\right)\left(b^3 + c^3 + abc\right)\left(c^3 + a^3 + abc\right)$. However, expanding $D$ yields a degree-9 polynomial in three variables with $\sim 55$ (via $\binom{9+3-1}{3-1} \approx 55$) monomials, rendering sum-of-squares or Fourier-Motzkin methods infeasible. Even if somehow the denominator are manually cleared, `nlinarith` can only handle (1) linear combinations of monomials (via `linarith`), (2) quadratic forms (by introducing auxiliary square variables and then linearizing),

| Error | DeepSeek-SFT | DeepSeek-RL | Goedel | STP | Kimina |
|---|---|---|---|---|---|
| Redundant Hypothesis | 18.0% | 34.0% | 27.0% | 24.0% | 36.0% |
| Incomplete Proof | 77.0% | 62.0% | 86.0% | 44.0% | 93.0% |
| Inabilities for Inequality | 8.0% | 13.0% | 20.0% | 1.0% | 20.0% |
| Misuse of Auto tactics | 62.0% | 65.0% | 78.0% | 74.0% | 43.0% |

Table 8: Percentage of different Lean4 error patterns in LLM-based provers.

and (3) simple monotonicity lemmas (*e.g.*, if $0 < x \le y \implies \frac{1}{x} \ge \frac{1}{y}$), but only after the provers normalize the goal via `ring` or `field` first.

**Redundant Hypothesis Introduction.** A common error in current LLM-based theorem provers arises from introducing structurally redundant hypotheses. While these do not inherently cause logical errors, they obscure the proof's underlying logic and reduce readability. For example, in the `aime_all_2005_II_1` proof (Table 11), the unnecessary use of `revert` followed by `reintro` exemplifies this issue. These tactics are designed to generalize variables or hypotheses—a technique critical for inductive proofs or hypothesis strengthening. However, in this case: (1) no inductive reasoning requires generalization, (2) the variables n, hn, and h already exist in the context and can be directly used. Thus, the tactic `revert` is redundant and can be removed to simplify the proof.

**Incomplete Proof.** Another common failure mode for for LLM-based provers is generating unfinished proof attempts that leave critical subgoals unresolved or rely on placeholder tactics without justifying intermediate steps. For example, in the proof sketch for `DEMIMathAnalysis_50` (Table 11), which aims to show $\lim_{n\to\infty} \sqrt{n} \cdot \int_{-\infty}^{\infty} \frac{1}{(1+x^2)^n} dx = \sqrt{\pi}$, the prover terminates prematurely after a few tactic calls that: (1) fail to justify interchanging the limit and integral and (2) fail to establish bounds on the integrand's tail decay. The flawed proof begins with an unnecessary rewrite of `sqrt` and misapplies monotonicity lemmas like `integral_mono_on` without verifying domination or integrability conditions required for the Dominated Convergence Theorem (Wikipedia contributors, 2024). Worse, tactics such as `tendsto_atTop_of_eventually_const` and `filter_upwards` trivialize tail behavior instead of rigorously addressing convergence. We hypothesize this error stems from short-sighted heuristic selection during language modeling of theorem provers: prioritizing tactics that maximize immediate log-probability or heuristic scores (*e.g.*, `gcongr`, `norm_num`, `simp`) over those advancing global proof progress. Such choices syntactically reshape goals while burying core challenges under shallow subgoals.

## 7 CONCLUDING REMARKS

FormalMATH is a new, extensive benchmark for evaluating LLMs' formal mathematical reasoning. It includes 5,560 formally verified Lean4 statements, covering topics from high-school Olympiads to undergraduate studies. We developed a human-in-the-loop autoformalization pipeline to create FormalMATH. This process uses specialized LLMs for initial Lean4 formalization, multi-LLM semantic verification to maintain fidelity to the original problems, and a negation-based disproof strategy to filter invalid statements. This significantly reduces manual review effort while achieving a 72.09% pre-verification preservation rate. Our evaluation of existing LLM-based theorem provers on FormalMATH shows considerable limitations. The best models achieve modest success, with the top performer reaching only 16.46% accuracy. The analysis also reveals strong domain biases: models perform better in areas like algebra but struggle in others, such as calculus. Furthermore, our findings suggest an over-reliance on simplified automation tactics and, surprisingly, a negative effect of natural-language solution guidance on proof success in CoT scenarios. These results underscore the difficulty of the FormalMATH benchmark and present key open problems for improving the robustness, generalizability, and reasoning complexity of automatic theorem provers.

ICLR PAPER CHECKLIST

1. **Ethics Statement**

   **Answer:** This research fully adheres to the ICLR Code of Ethics. The study does not involve human subjects or the use of personal or sensitive data. All datasets and code utilized and released conform to their respective licenses and terms of use. If any annotation tasks were involved, annotators merely labeled mathematical problems, and no risk or personal information was associated with their participation. The contributions in this work are foundational and do not raise issues related to fairness, privacy, security, or potential misuse. We confirm that all ethical considerations have been thoroughly addressed.

2. **Reproducibility Statement**

   **Answer:** We are committed to making our work easily reproducible. All essential details required to replicate our main experimental results—including data access, experimental setup, model configurations, and evaluation metrics—are provided either on the designated project page. Released code and datasets come with clear instructions to reproduce both our proposed method and baseline experiments. We specify all training and test parameters as well as compute resource requirements. Users can follow our documentation and scripts to faithfully reproduce the results, ensuring transparency and scientific rigor.

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

## A  THE FORMALMATH DATASET

The concrete dataset of FormalMATH, including FormalMATH-All and its simpler subset (*i.e.*, FormalMATH-Lite), can be found at:
https://anonymous.4open.science/r/formalmath_rebuttal-3CED

## B  DATA SOURCES

Table 9 presents the sources of the natural language datasets used in the FormalMATH project.

| Dataset | Level | #Domains | Size | #S.Formal |
|---|---|---|---|---|
| Omni-math (Gao et al., 2024) | High School Olympiad | 9 | 4.43k | 1,210 |
| Numina-Olympiad | High School Olympiad | 10 | 11.8k | 2,409 |
| AIME-Math | High School Olympiad | 7 | 934 | 371 |
| BlueMO (Zhang et al., 2024) | High School Olympiad | 8 | 3,024 | 1,099 |
| U-Math (Chernyshev et al., 2024) | Undergraduate | 6 | 1,100 | 358 |
| Hardmath (Fan et al., 2024) | Undergraduate | 3 | 1,060 | 67 |
| DEMIMATH (Hata, 2016) | Undergraduate | 8 | 88 | 46 |

Table 9: The sources of the datasets in FormalMATH. "#Domains" denotes the number of domains in the dataset. "#S.Formal" denotes the number of samples in FormalMATH that are formalized from the dataset.

## C  EXAMPLES OF STATEMENTS IN FORMALMATH

---

**Example B.1: The 27th derivative for $y = 2 \cdot x^2 \cdot \sin(x)$**

**u-math_915**

```
Find the derivative of the 27th order y^(27) for a function y = 2 · x² · sin(x). Prove that the answer
    is: y^(27) = 1404 · cos(x) − 2 · x² · cos(x) − 108 · x · sin(x)
```

**Autoformalization:**

```
import Mathlib

open Real Set
open scoped BigOperators

theorem u_math_915 {f : ℝ → ℝ} (hf : f = λ x => 2 * x ^ 2 * sin x):
    iteratedDeriv 27 f = λ x => 1404 * cos x − 2 * x ^ 2 * cos x − 108 * x * sin x := by
```

---

---

**Example B.2: The analytical formula for $I(x)$ — A Nontrivial Polynomial-Exponential Integral**

**hardmath_569**

```
Consider the integral
```
$I(x) = \int_{-0.3}^{0.6} \left(1.2t^5 + 2.4t^4 + 1.7t^3 - 0.6\,\mathrm{atan}(t) + 1.3\right) e^{x\left(1.2t^5 - 0.8t^4 - 2.7t^3 - 2.6t^2\right)}\,dt$
```
Develop an analytical formula for I(x) that is accurate as x → ∞.
Prove that the answer is:
```
$I(x) \approx \frac{\sqrt{65}\sqrt{\pi}\sqrt{\frac{1}{x}}}{10}$.

**Autoformalization:**

**import** Mathlib

**open** Real

**theorem** hardmath_569 $(f : \mathbb{R} \to \mathbb{R})$ $(\mathtt{hf} : \mathtt{f} = \mathbf{fun}\ \mathtt{x} => \int \mathtt{t}\ \mathbf{in}\ (-0.3)..0.6, (1.2 * \mathtt{t}\hat{}5 + 2.4 * \mathtt{t}\hat{}4 + 1.7 * \mathtt{t}\hat{}3 - 0.6 *$
$\qquad \mathtt{arctan}\ \mathtt{t} + 1.3) * \exp(\mathtt{x} * (1.2 * \mathtt{t}\hat{}5 - 0.8 * \mathtt{t}\hat{}4 - 2.7 * \mathtt{t}\hat{}3 - 2.6 * \mathtt{t}\hat{}2)))$ :

$\forall \varepsilon > 0, \exists\, \mathtt{x}, \forall\, \mathtt{y}, \mathtt{y} > \mathtt{x} \to |\mathtt{f}\ \mathtt{y} - (\mathtt{Real.sqrt}\ 65 * \mathtt{Real.sqrt}\ \pi * \mathtt{Real.sqrt}\ (1\,/\,\mathtt{y}))\,/\,10| < \varepsilon :=$ **by**

---

**Example B.3: Find all positive integer solutions to a nonlinear divisibility equation**

**omni_theorem_4136**

```
Determine all pairs $(x, y)$ of positive integers such that $x^2 y + x + y$ is divisible by $xy^2 + y + 7$.
```
```
Prove that the answer is: $(x, y) = (11, 1), (49, 1), (7t^2, 7t)$, $t$ is an interge
```

**Autoformalization:**

**import** Mathlib

**open** Real

**open** scoped BigOperators

**theorem** omni_theorem_4136 : $\forall\, \mathtt{x}\ \mathtt{y} : \mathbb{N}, \mathtt{x} > 0 \wedge \mathtt{y} > 0 \to ((\mathtt{x}\hat{}2 * \mathtt{y} + \mathtt{x} + \mathtt{y})\,\%\,(\mathtt{x} * \mathtt{y}\hat{}2 + \mathtt{y} + 7) = 0 \leftrightarrow$

$(\mathtt{x} = 11 \wedge \mathtt{y} = 1) \vee (\mathtt{x} = 49 \wedge \mathtt{y} = 1) \vee \exists\, \mathtt{t} : \mathbb{Z}, \mathtt{x} = 7 * \mathtt{t}\hat{}2 \wedge \mathtt{y} = 7 * \mathtt{t}) :=$ **by**

---

# D  THE ERROR TYPES OF OUR AUTOFORMALIZATION PIPELINE

| Error Category | Percentage (%) |
|---|---|
| Condition Error | 83.3 |
| Expression Error (Lean Syntax) | 9.6 |
| Definition Error (No Mathematical Meaning) | 3.3 |
| Domain Error | 1.7 |
| Propositional Logic Error | 0.9 |
| Lack of Geometric Background | 0.6 |
| Condition Redundancy | 0.5 |
| Algebraic Expression Error | 0.2 |

Table 10: Error Classification Statistics (%)

# E  DOMAIN DISTRIBUTION OF FORMALMATH-LITE

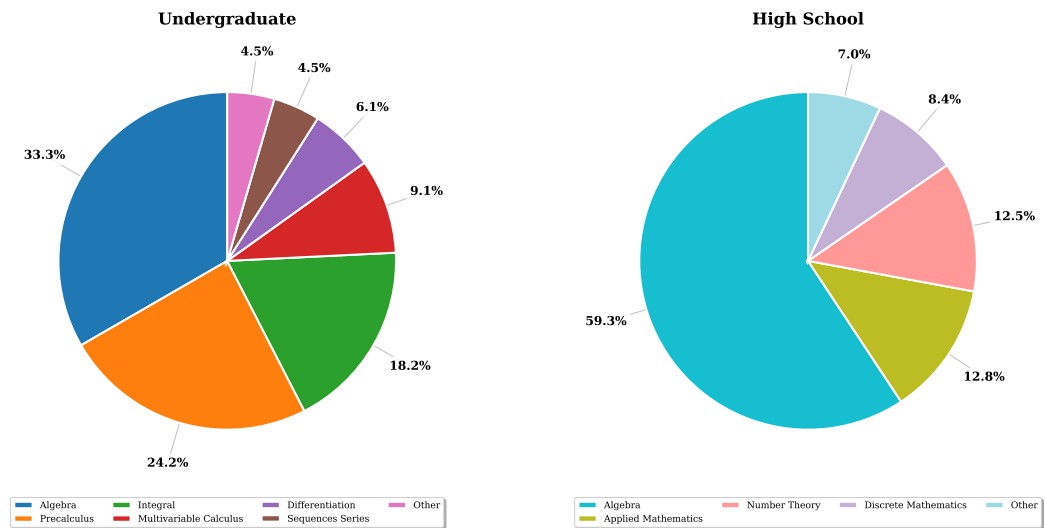

Figure 4: The distribution of mathematical domains in our FormalMATH-Lite dataset.

# F  TYPICAL ERRORS IN STATEMENT AUTOFORMALIZATION

## F.1  ERRORS IN DEFINITION

Definition Errors mainly occur in the failure of semantically corresponding the mathematical entities in autoformalized statements one-to-one with those in the refined statement. For instance, in a geometry problem, if the defined entities do not pertain to geometric concepts such as points, lines, and angles, it indicates a lack of domain relevance.

Mismatched Variable type is also a common issue. This type of error can lead to failure when applying tactics that are specific to a particular variable type.

Furthermore, the defined variables may lack actual mathematical significance so that the statement cannot pass the compiler.

---

**Example E.1: Errors in Definition: Lack of Specific Mathematical Background (Geometry)**

**omni_theorem_2599**

Points $A, B, C, D$ lie on a circle in that order such that $\frac{AB}{BC} = \frac{DA}{CD}$. If $AC = 3$ and $BD = BC = 4$, find $AD$.
Prove that the answer is: $\frac{3}{2}$

**Autoformalization:**

```
import Mathlib

open EuclideanGeometry Real
open scoped BigOperators

theorem omni_theorem_2599
  {A B C D : EuclideanSpace ℝ (Fin 2)}
  (h₀ : ∃ O, dist A O = dist B O ∧ dist B O = dist C O ∧ dist C O = dist D O)
  (h₁ : dist A B / dist B C = dist D A / dist C D)
  (h₂ : dist A C = 3)
  (h₃ : dist B D = 4)
  (h₄ : dist B C = 4) :
  dist A D = 3 / 2 := by sorry
```

---

The model's formalization is deficient because it only specifies the lengths of line segments $AB$, $BC$, $CD$, and $DA$, without explicitly stating that points $A$, $B$, $C$, and $D$ are in that order along circle O. This omission is critical as only segment lengths is insufficient to uniquely define a quadrilateral

inscribed in a circle. Different orderings of these points on the circle, even with the same segment lengths.

---

**Example E.2: Errors in Definition: Mismatched variable Type**

**omni_theorem_66**

Determine whether or not there exist two distinct sets $A$ and $B$, each consisting of at most $2011^2$ positive integers, such that for every $x$ with $0 < x < 1$, the following inequality holds: $\left| \sum_{a \in A} x^a - \sum_{b \in B} x^b \right| < (1 - x)^{2011}$.
Prove that the answer is Yes.

**Autoformalization:**

```
import Mathlib

theorem omni_theorem_66 :
  ∃ A B : Finset ℕ,
  A ≠ B ∧
  A.card ≤ 2011^2 ∧
  B.card ≤ 2011^2 ∧
  (∀ x ∈ Set.Ioo 0 1, |Σ a ∈ A, (x : ℝ)^a − Σ b ∈ B, (x : ℝ)^b| < (1 − x)^2011) := by sorry
```

---

While positive integer mappings and positive integers m and n were required, the autoformalization mistakenly used natural numbers instead.

---

**Example E.3: Errors in Definition: Meaningless Definition in Mathematical Context**

**omni_theorem_1024**

For a positive integer $n$, denote by $\tau(n)$ the number of positive integer divisors of $n$, and denote by $\phi(n)$ the number of positive integers that are less than or equal to $n$ and relatively prime to $n$. Call a positive integer $n$ good if $\varphi(n) + 4\tau(n) = n$. For example, the number 44 is good because $\varphi(44) + 4\tau(44) = 44$. Find the sum of all good positive integers $n$.
Prove that the answer is: 172

**Autoformalization:**

```
import Mathlib

open Finset

def isGood (n : ℕ) := n.totient + 4 * n.divisors.card = n

theorem omni_theorem_1024 :
  Σ n ∈ {n | 0 < n ∧ isGood n}, n = 172 := by sorry
```

---

Euler's totient function, denoted as $\phi(n)$ or in Lean4 as `n.totient`, counts the number of positive integers less than or equal to $n$ that are relatively prime to $n$. For $n = 0$, the function `n.totient` is not defined, which results in isGood n being undefined.

## F.2 ERRORS IN EXPRESSIONS

This type of errors include incorrect arithmetic or logical operations, improper use of variables, or incorrect application of mathematical or logical rules.

---

**Example E.4: Errors in Proposition Construction (Wrong Position of Existential Qualifier)**

**omni_theorem_4**

```
Let p be a prime number. Prove the following theorem of Euler: the equation p = x² + 3y² has a
    solution with x, y ∈ ℤ if and only if p = 3 or p ≡ 1 (mod 3). (You may use the fact that the
    ring of integers of ℚ(√−3) is a principal ideal domain.)
Prove that the answer is: The equation p = x² + 3y² has a solution with x, y ∈ ℤ if and only if p =
    3 or p \equiv 1 \pmod{3}
```

**Autoformalization:**

```
import Mathlib

open Real
open scoped BigOperators

theorem omni_theorem_401 {p : ℕ} (hp : Nat.Prime p) :
    ∃ x y : ℤ, p = x^2 + 3*y^2 ↔ p = 3 ∨ p % 3 = 1 := by sorry
```

In the Lean4 code, the existential quantifier is misplaced. $\exists x, y \in \mathbb{Z}$ should quantify the entire left-hand side of the biconditional. Instead of asserting that the existence of a solution $(x, y)$ is equivalent to the condition $p = 3 \lor p \equiv 1 \pmod 3$, the incorrect code unintentionally suggests the existence of $x$ and $y$ is linked to the entire biconditional statement being true. the existential quantifier should be correctly scoped only to the left-hand side of the biconditional, as in the corrected form:

$$(\exists x, y \in \mathbb{Z}, p = x^2 + 3y^2) \leftrightarrow (p = 3 \lor p \equiv 1 \pmod 3),$$

which ensures that the quantifier only applies to the equation $p = x^2 + 3y^2$.

---

**Example E.5: Errors in Expression: Misleading Scope of the Universal Quantifier**

**omni_theorem_219**

```
Find, with proof, all nonconstant polynomials P(x) with real coefficients such that, for all
    nonzero real numbers z with P(z) ≠ 0 and P (1/z) ≠ 0, we have

\frac{1}{P(z)}+\frac{1}{P\left(\frac{1}{z}\right)}=z+\frac{1}{z}

Prove that the answer is:

P(x) = \frac{x(x^{4k+2}+1)}{x^{2}+1} \quad \text{or} \quad P(x) = \frac{x(1−x^{4k})}{x^{2}+1}
```

**Autoformalization:**

```
import Mathlib

open Fintype Set Real Ideal Polynomial
open scoped BigOperators

theorem omni_theorem_219 {P : Polynomial ℝ} (hP : P.degree > 0) :
    ∀ z : ℝ, z ≠ 0 → P.eval z ≠ 0 → P.eval (1/z) ≠ 0 →
    1/(P.eval z) + 1/(P.eval (1/z)) = z + 1/z ↔
    ∃ k : ℕ, P = X * (X^(4*k + 2) + 1)/(X^2 + 1) ∨
    P = X * (1 − X^(4*k))/(X^2 + 1) := by sorry
```

The problem is one of logical scope and intended semantic meaning. While Lean4's type system is powerful and can often handle somewhat flexible syntax, the initial phrasing can lead to a misinterpretation of the theorem's claim. While Lean4 might parse this code without immediate syntax errors due to the right-associativity of implication, this placement leads to a misinterpretation of the theorem's intended logical structure and meaning. The original code is effectively parsed as if it were written:

---

**Example E.6: Logical Parse in Original Autoformalized Statement**

**omni_theorem_219-autoformalization**

$\forall$ z : $\mathbb{R}$, (z $\neq$ 0 $\rightarrow$ (P.eval z $\neq$ 0 $\rightarrow$ (P.eval (1/z) $\neq$ 0 $\rightarrow$
  (1/(P.eval z) + 1/(P.eval (1/z)) = z + 1/z $\leftrightarrow$
  $\exists$ k : $\mathbb{N}$, P = X * (X^(4*k + 2) + 1)/(X^2 + 1) $\vee$
  P = X * (1 − X^(4*k))/(X^2 + 1) ))))

---

The theorem should state: A nonconstant polynomial $P(x)$ satisfies the property that for all relevant nonzero $z$, the equation holds if and only if $P(x)$ takes one of the specified forms. To accurately reflect the intended meaning and correct the quantifier placement, we must use parentheses to explicitly define the scope of the universal quantifier.

---

**Example E.7: Enhanced Autoformalized Statement**

**omni_theorem_219-autoformalization**

```
theorem omni\_theorem\_219 \{P : Polynomial ℝ\} (hP : P.degree > 0) : \\
  (∀ z : ℝ, z ≠ 0 → P.eval z ≠ 0 → P.eval (1/z) ≠ 0 →
   1/(P.eval z) + 1/(P.eval (1/z)) = z + 1/z) ↔
  ∃ k : ℕ, P = X * (X^(4*k + 2) + 1)/(X^2 + 1) ∨
  P = X * (1 − X^(4*k))/(X^2 + 1) := by sorry
```

---

## F.3   ERRORS IN CONSTRAINT CONDITION

The content of the constraints may not match the requirements stated in the problem or be left out. We also notice that constraints that are not mentioned in the original problem statement may be added during formalization.

---

**Example E.8: Errors in Constraint Condition: Mismatch with known conditions**

**omni_theorem_748:**

```
Let f(x) = x² + ax + b and g(x) = x² + cx + d be two distinct real polynomials such that the x
    −coordinate of the vertex of f is a root of g, the x−coordinate of the vertex of g is a root of
    f and both f and g have the same minimum value. If the graphs of the two polynomials
    intersect at the point (2012, − 2012), what is the value of a + c ?
Prove that the answer is: −8048
```

**Autoformalization:**

```
import Mathlib

theorem omni_theorem_748 {a b c d : ℝ} (f g : ℝ → ℝ)
  (hf : f = fun x => x^2 + a * x + b) (hg : g = fun x => x^2 + c * x + d)
  (hfg : f ≠ g) (hvertex : ∃ x, x = −a / 2 ∧ g x = 0) (hvertex' : ∃ x, x = −c / 2 ∧ f x = 0)
  (hmin : ∃ y, ∀ x, f x ≥ y ∧ g x ≥ y) (hintersect : f 2012 = −2012 ∧ g 2012 = −2012) :
  a + c = −8048 := by sorry
```

---

In Autoformalization, `hmin`: $\exists y, \forall x, f(x) \geq y \wedge g(x) \geq y$ means that $f$ and $g$ have a common lower bound, rather than: $f$ and $g$ have the same minimum value.

---

**Example E.8: Errors in Constraint Condition: Lack of a key condition**

**omni_theorem_10:**

```
Given 30 students such that each student has at most 5 friends and for every 5 students there is a
    pair of students that are not friends, determine the maximum k such that for all such
    possible configurations, there exists k students who are all not friends.
Prove that the answer is: 6
```

**Autoformalization:**

**import** Mathlib

**open** Finset

**theorem** omni_theorem_10 (n : ℕ) (hn : n = 30) (f : Fin n → Fin n → Prop)
(hf : ∀ i j, f i j ↔ f j i) (hfn : ∀ s : Finset (Fin n), s.card = 5 →
∃ i ∈ s, ∃ j ∈ s, i ≠ j ∧ ¬f i j) :
IsGreatest {k | ∃ t : Finset (Fin n), t.card = k ∧ ∀ i ∈ t, ∀ j ∈ t, i ≠ j → ¬f i j} 6 := **by** sorry

---

The formalization fails to represent the condition "Each student has at most 5 friends". This crucial condition is necessary for accurately capturing the problem's constraints, and its absence may lead to a solution space that does not match the requirements of the original problem. Without this condition, the formalized statement does not fully reflect the situation described in the refined statement, potentially resulting in an incorrect or incomplete proof for determining the maximum number k of students who are all not friends.

---

**Example E.9: Errors in Constraint Condition: Redundant conditions**

**omni_theorem_1245**

```
The fraction 1/2015 has a unique "(restricted) partial fraction decomposition" of the form
    1/2015 = a/5 + b/13 + c/31 where a, b, c are integers with 0 ≤ a < 5 and 0 ≤ b < 13. Find a + b.
 Prove that the answer is: 14
```

**Autoformalization:**

**import** Mathlib

**theorem** omni_theorem_1245 {a b c : ℤ} (ha : 0 ≤ a ∧ a < 5) (hb : 0 ≤ b ∧ b < 13)
(hc : 0 ≤ c ∧ c < 31) (h : 1 / 2015 = a / 5 + b / 13 + c / 31) : a + b = 14 := **by** sorry

---

In autoformalization, the model introduced a spurious condition: (hc :   0 ≤ c ∧ c < 31).

## F.4 ERRORS IN PROOF GOALS

The stated proof goals may be incorrect. They could be misaligned with the original problem statement, overly simplified, too general or even omitted.

---

**Example E.10: Errors in Proof Goals: Incorrect Proposition Logic**

**omni_theorem_1362**

```
The Fibonacci sequence is defined as follows: F₀ = 0, F₁ = 1, and Fₙ = Fₙ₋₁ + Fₙ₋₂ for all
    integers n ≥ 2. Find the smallest positive integer m such that Fₘ ≡ 0(mod 127) and
    Fₘ₊₁ ≡ 1(mod 127).
Prove that the answer is: 256
```

**Autoformalization:**

```
import Mathlib

def fib (n : ℕ) : ℕ := match n with
| 0 => 0
| 1 => 1
| n + 2 => fib (n + 1) + fib n

theorem omni_theorem_1362 {m : ℕ} (hm : m > 0) :
  fib m ≡ 0 [MOD 127] ∧ fib (m + 1) ≡ 1 [MOD 127] ↔ m = 256 := by sorry
```

---

The fundamental error in this autoformalization lies in framing the problem as an equivalence verification. The autoformalization, by using $\leftrightarrow m = 256$, transforms the problem into a claim that the given congruences hold if and only if $m$ is exactly 256. The correct approach to formalize the original problem would involve demonstrating that The value $m = 256$ satisfies the conditions:fib256 $\equiv 0 \pmod{127} \wedge$ fib$(256 + 1) \equiv 1 \pmod{127}$,and that no smaller positive integer $m' < 256$ also satisfies these conditions.

---

**Example E.11: Errors in Proof Goals: Incomplete Goal**

**olymidrefbase_551**

```
Let a, b, c, d, e, f be non−negative real numbers satisfying a + b + c + d + e + f = 6. Find the
    maximal possible value of  a b c+b c d+c d e+d e f+e f a+f a b  and determine all 6−tuples
    (a, b, c, d, e, f) for which this maximal value is achieved. Answer: 8. Prove that the answer is: 8
```

**Autoformalization:**

```
import Mathlib open Real theorem olymid_ref_base_551:  IsGreatest {x | ∃ a b c d e f : ℝ, a ≥ 0 ∧ b ≥
    0 ∧ c ≥ 0 ∧ d ≥ 0 ∧ e ≥ 0 ∧ f ≥ 0 ∧ a + b + c + d + e + f = 6 ∧ x = a * b * c + b * c * d + c * d * e + d * e * f +
    e * f * a + f * a * b} 8 := by sorry
```

---

The core issue lies in how the autoformalization treats the problem's objective – finding the maximal possible value – and the request to determine all 6-tuples that achieve this maximum. The original problem requires the solver to not only find the maximum value but also to characterize the set of inputs that lead to this maximum. The provided autoformalization using IsGreatest completely omits any formalization of the requirement to determine all 6-tuples. It focuses solely on verifying the maximal value (8).

## G  PROMPT FOR SEMANTIC VERIFICATION

To more effectively evaluate the consistency between natural language mathematics problems and their Lean4 formalizations, we adopted an LLMs group voting approach to filter entries with the same semantics. The prompt provided to the five LLMs is as follows:

---

**Example F.1: Prompt for Semantic Verification**

You are an expert in formalizing natural language into lean.
You are given a natural language statement and a lean statement.
You should judge the equivalence between the natural language statement and the lean statement by the following workflow:
1. You should back-translate the lean statement into English.
2. You should check if the back-translated statement is equivalent to the natural language statement.
3. If they are equivalent, you should return True.
4. Otherwise, you should return False.
Here is the natural language statement:
{refined_statement}
Here is the lean statement:
{lean_statement}
You must remember : Return True or False directly. Accept only True/False in answer.

---

## H  PROMPT FOR DOMAIN CLASSIFICATION

---

**Example G.1: Prompt for Domain Classification**

```
# CONTEXT #
I am a teacher, and I have some high-level math problems.
I want to categorize the domain of these math problems.

# OBJECTIVE #
A. Summarize the math problem in a brief sentence, describing the concepts involved in
the math problem.
B. Categorize the math problem into specific mathematical domains. Please provide a
classification chain, for example: Mathematics -> Applied Mathematics -> Probability
-> Combinations. The following is a basic classification framework in the field of
mathematics.
<math domains>
Mathematics
|
|-- Applied Mathematics
| |-- Math Word Problems
| |-- Statistics
| |-- Mathematical Statistics
| |-- Probability
| |-- Counting Methods
| |-- Permutations
| |-- Combinations
|
|-- Algebra
| |-- Prealgebra
| |-- Integers
| |-- Fractions
| |-- Decimals
| |-- Simple Equations
| |-- Algebra
| |-- Algebraic Expressions
| |-- Equations and Inequalities
| |-- Factoring
| |-- Polynomial Operations
| |-- Intermediate Algebra
| |-- Quadratic Functions
| |-- Exponential Functions
| |-- Logarithmic Functions
| |-- Complex Numbers
| |-- Linear Algebra
| |-- Vectors
| |-- Matrices
| |-- Determinants
| |-- Linear Transformations
| |-- Abstract Algebra
| |-- Group Theory
| |-- Ring Theory
| |-- Field Theory
```

---

```
1188    |
1189    |-- Geometry
1190    | |-- Plane Geometry
1191    | |-- Polygons
        | |-- Angles
1192    | |-- Area
1193    | |-- Triangulations
        | |-- Perimeter
1194    | |-- Solid Geometry
1195    | |-- 3D Shapes
        | |-- Volume
1196    | |-- Surface Area
1197    | |-- Differential Geometry
        | |-- Curvature
1198    | |-- Manifolds
1199    | |-- Geodesics
        | |-- Non-Euclidean Geometry
1200    | |-- Spherical Geometry
1201    | |-- Hyperbolic Geometry
        |
1202    |-- Number Theory
1203    | |-- Prime Numbers
        | |-- Factorization
1204    | |-- Congruences
1205    | |-- Greatest Common Divisors (GCD)
        | |-- Least Common Multiples (LCM)
1206    |
1207    |-- Precalculus
        | |-- Functions
1208    | |-- Limits
1209    | |-- Trigonometric Functions
        |
1210    |-- Calculus
1211    | |-- Differential Calculus
        | |-- Derivatives
1212    | |-- Applications of Derivatives
1213    | |-- Related Rates
        | |-- Integral Calculus
1214    | |-- Integrals
1215    | |-- Applications of Integrals
        | |-- Techniques of Integration
1216    | |-- Single-variable
1217    | |-- Multi-variable
        |
1218    |-- Differential Equations
1219    | |-- Ordinary Differential Equations (ODEs)
        | |-- Partial Differential Equations (PDEs)
1220    |
1221    |-- Discrete Mathematics
        |-- Graph Theory
1222    |-- Combinatorics
1223    |-- Logic
        |-- Algorithms
1224    </math domains>
1225
        # STYLE #
1226    Data report.
1227
        # TONE #
1228    Professional, scientific.
1229
        # AUDIENCE #
1230    Students.  Enable them to better understand the domain and difficulty of the math
1231    problems.
1232    # RESPONSE: MARKDOWN REPORT # ## Summarization [Summarize the math problem in a brief
1233    paragraph.]  ## Math domains [Categorize the math problem into specific mathematical
        domains, including major domains and subdomains.]`
1234    # ATTENTION # - The math problem can be categorized into multiple domains, but no more
1235    than three.  Separate the classification chains with semicolons(;).
        - Your classification MUST fall under one of the aforementioned subfields; if it
1236    really does not fit, please add "Other" to the corresponding branch.  For example:
1237    Mathematics -> Algebra -> Intermediate Algebra -> Other.  Only the LAST NODE is allowed
        to be "Other"; the preceding nodes must strictly conform to the existing framework.
1238    - The math domain must conform to a format of classification chain, like "Mathematics
1239    -> Applied Mathematics -> Probability -> Combinations".
        - Add "=== report over ===" at the end of the report.
1240
1241
```

<example math problem>

[Question]:
If $\frac{1}{9} + \frac{1}{18} = \frac{1}{square}$, what is the number that replaces the *square* to make the equation true?
[Solution]:
We simplify the left side and express it as a fraction with numerator 1: $\frac{1}{9} + \frac{1}{18} = \frac{2}{18} + \frac{1}{18} = \frac{3}{18} = \frac{1}{6}$. Therefore, the number that replaces the *square* is 6.
[Source]: 2010 Pascal:

</example math problem>
## Summarization The problem requires finding a value that makes the equation $\frac{1}{9} + \frac{1}{18} = \frac{1}{square}$. This involves adding two fractions and determining the equivalent fraction.
## Math domains Mathematics -> Algebra -> Prealgebra -> Fractions;

=== report over ===

</example math problem>
[Question]:
Let $\mathcal{P}$ be a convex polygon with $n$ sides, $n \geq 3$. Any set of $n - 3$ diagonals of $\mathcal{P}$ that do not intersect in the interior of the polygon determine a triangulation of $\mathcal{P}$ into $n - 2$ triangles. If $\mathcal{P}$ is regular and there is a triangulation of $\mathcal{P}$ consisting of only isosceles triangles, find all the possible values of $n$.
[Solution]:
We label the vertices of $\mathcal{P}$ as $P_0, P_1, P_2, \ldots, P_n$. Consider a diagonal $d = \overline{P_a P_{a+k}}$, $k \leq n/2$ in the triangulation. We show that $k$ must have the form $2^m$ for some nonnegative integer $m$. This diagonal partitions $\mathcal{P}$ into two regions $\mathcal{Q}, \mathcal{R}$, and is the side of an isosceles triangle in both regions. Without loss of generality suppose the area of $Q$ is less than the area of $R$ (so the center of $P$ does not lie in the interior of $Q$); it follows that the lengths of the edges and diagonals in $Q$ are all smaller than $d$. Thus $d$ must the be the base of the isosceles triangle in $Q$, from which it follows that the isosceles triangle is $\triangle P_a P_{a+k/2} P_{a+k}$, and so $2|k$. Repeating this process on the legs of isosceles triangle $(\overline{P_a P_{a+k/2}}, \overline{P_{a+k} P_{a+k/2}})$, it follows that $k = 2^m$ for some positive integer $m$ (if we allow degeneracy, then we can also let $m = 0$). Now take the isosceles triangle $P_x P_y P_z$, $0 \leq x < y < z < n$ in the triangulation that contains the center of $\mathcal{P}$ in its interior; if a diagonal passes through the center, select either of the isosceles triangles with that diagonal as an edge. Without loss of generality, suppose $P_x P_y = P_y P_z$. From our previous result, it follows that there are $2^a$ edges of $P$ on the minor arcs of $P_x P_y$, $P_y P_z$ and $2^b$ edges of $P$ on the minor arc of $P_z P_x$, for positive integers $a, b$. Therefore, we can write
$$n = 2 \cdot 2^a + 2^b = 2^{a+1} + 2^b,$$
so $n$ must be the sum of two powers of $2$. We now claim that this condition is sufficient. Suppose without loss of generality that $a + 1 \geq b$; then we rewrite this as
$$n = 2^b(2^{a-b+1} + 1).$$
Lemma 1: All regular polygons with $n = 2^k + 1$ or $n = 4$ have triangulations that meet the conditions. By induction, it follows that we can cover all the desired $n$. For $n = 3, 4$, this is trivial. For $k > 1$, we construct the diagonals of equal length $\overline{P_0 P_{2k-1}}$ and $\overline{P_{2k-1+1} P_0}$. This partitions $\mathcal{P}$ into 3 regions: an isosceles $\triangle P_0 P_{2k-1} P_{2k-1+1}$, and two other regions. For these two regions, we can recursively construct the isosceles triangles defined above in the second paragraph. It follows that we have constructed $2(2^{k-1} - 1) + (1) = 2^k - 1 = n - 2$ isosceles triangles with non-intersecting diagonals, as desired.
Lemma 2: If a regular polygon with $n$ sides has a working triangulation, then the regular polygon with $2n$ sides also has a triangulation that meets the conditions. We construct the diagonals $\overline{P_0 P_2}$, $\overline{P_2 P_4}$, $\ldots \overline{P_{2n-2} P_0}$. This partitions $\mathcal{P}$ into $n$ isosceles triangles of the form $\triangle P_{2k} P_{2k+1} P_{2k+2}$, as well as a central regular polygon with $n$ sides. However, we know that there exists a triangulation for the $n$-sided polygon that yields $n - 2$ isosceles triangles. Thus, we have created $(n) + (n - 2) = 2n - 2$ isosceles triangles with non-intersecting diagonals, as desired. In summary, the answer is all $n$ that can be written in the form $2^{a+1} + 2^b, a, b \geq 0$. Alternatively, this condition can be expressed as either $n = 2^k$, $k \geq 2$ (this is the case when $a + 1 = b$) or $n$ is the sum of two distinct powers of $2$, where $1 = 2^0$ is considered a power of $2$.
[Source]:
USAMO 2008
</example math problem>

## Summarization
The problem asks for the possible values of $n$ for a regular $n$-sided polygon that can be completely triangulated into isosceles triangles using non-intersecting diagonals. The solution involves analyzing the properties of the diagonals forming isosceles triangles and deducing that $n$ can be expressed in terms of powers of $2$.
## Math domains
Mathematics -> Geometry -> Plane Geometry -> Polygons;

=== report over ===

# I PROMPTS FOR THEOREM PROVERS

## I.1 PROMPT FOR VANILLA GENERATION

---

**Example H.1: Prompt for Vanilla Generation**

Complete the following Lean4 code:
```lean4
import Mathlib

theorem omni_theorem_2669
```
$(x : \mathbb{Z})\ (hx : x = 2018) : x^2 + 2 * x - x * (x + 1) = 2018 := by$

---

## I.2 PROMPT FOR CoT GENERATION

---

**Example H.2: Prompt for CoT Generation**

Complete the following Lean4 code with explanatory comments preceding each line of code:
```lean4
import Mathlib

theorem omni_theorem_2669
```
$(x : \mathbb{Z})\ (hx : x = 2018) : x^2 + 2 * x - x * (x + 1) = 2018 := by$

---

## I.3 PROMPT FOR NL-AUGMENTED CoT

---

**Example H.3: Prompt for CoT with Natural Solution Generation**

Complete the following Lean4 code with explanatory comments preceding each line of code:
```lean4
import Mathlib
open Finset
theorem omni_theorem_4199 :
```
$\exists n \in \mathbb{N}, \{s : \text{Finset}\mathbb{N} \mid s.\text{card} = 2017 \wedge \sum_{i \in s} i^2 = n\}.\text{ncard} \geq 2017 := \text{by}$
/-To determine if there exists a number $n$ that can be expressed as the sum of 2017 perfect squares in at least 2017 distinct ways, we consider the properties and combinations of perfect squares.
### Step 1: Understanding the Problem
The problem asks us to express a number $n$ as the sum of 2017 perfect squares, $n = a_1^2 + a_2^2 + \cdots + a_{2017}^2$, where $a_i$ are integers. Moreover, this can be done in at least 2017 different ways, meaning there are at least 2017 distinct sets of such integers.
### Step 2: Exploring Perfect Squares
Perfect squares are non-negative numbers of the form $k^2$, where $k$ is an integer. To construct different sums, we need to evaluate how the combinations of these squares can vary and still yield distinct sums that equate to the same $n$.
### Step 3: Existence of Solutions
1. **Many Small Squares**: By choosing different arrangements of small perfect squares (like 0, 1, 4, 9, etc.), we can vary them freely since they don't drastically alter the cumulative sum quickly. For instance, using 0 is trivial as it adds nothing to sums; including or excluding it in varying positions introduces variety.
2. **Adjusting a Larger Value**: Consider including a larger square, say $(k + 1)^2$, and adjusting the rest of the terms accordingly. This diversity of combinations even with fixed values of $a_i = 0$ (i.e., not all contributing to sum) provides additional distinct setups.
### Step 4: Conclusion
Given the vast number of combinations possible with 2017 variables, it is feasible to achieve at least 2017 distinct sums since:
- Choosing different subsets of minimal contributions (e.g., many zeros and small numbers) can still lead to varying sums.
- Incremental adjustments in a few selections using larger squares or varied middle-range integers allow differential assembly leading to the target sum.
Thus, there is indeed a number $n$ that can be expressed as the sum of 2017 perfect squares in at least 2017 distinct ways.
Hence, the answer is:
Yes -/

---

## J PROMPT FOR ERROR PATTERN DIAGNOSIS

---

**Example I.1: Prompt for Error Pattern Diagnosis**

**Role:** Lean4 Error Pattern Analyst

**Input:** You will be provided with a list containing 5 Lean4 code snippets. Assume these snippets contain errors or represent incorrect usage patterns.

**Task:** Analyze all 5 snippets and identify the **common features or error patterns** present across them.

**Output:** Generate a list of concise strings describing these common features. Each string should be a short label for the pattern.

**Constraints:** * Focus *only* on identifying common features/errors across the provided 5 snippets. * Do **not** correct or modify the code. * Keep feature descriptions brief and informative (e.g., "Misuse of automated tactic", "Type mismatch in arguments", "Incorrect proof structure", "Syntax error in definition").

**Example Input Snippets (Conceptual):** [Lean4 Code Snippet 1 (Incorrect), ..., Lean4 Code Snippet 5 (Incorrect)],

**Example Output:** [ "Misuse of automated tactic": detailed reason, and exactly which problems (using problem id) make this fault. .... ] each feature should be mutually exclusive, and the features should cover all the common features of the code.
**Analyze the following 5 Lean4 code snippets:**

---

## K PROMPT FOR ERROR PATTERN CATEGORIZATION

---

**Example J.1: Prompt for Lean4 Proof Error Classification**

**Role:** Lean4 Code Classifier
**Task:** Classify the given Lean4 code snippet into one or more of the following categories based on the identified error patterns:

1. Improper usage of the automation tactics
2. Incomplete or Placeholder Proof Steps
3. Misuse of rewriting/simplification tactics
4. Inadequate handling of inequalities
5. Redundant hypothesis introductions

**Output Format:** Return a JSON object with the following structure:

{
"categories": ["category1", "category2", ...],
"confidence": [0.8, 0.7, ...], # Confidence scores for each category
"explanation": "Brief explanation of why these categories were chosen"
}
**Code to Classify:**

---

## L COMPUTE RESOURCES

Our experiments, including Pass@32 on FormalMATH-All and Pass@3200 on FormalMATH-Lite, require at least 8 NVIDIA H100 GPUs running for 2-3 days to generate outputs, followed by an additional 2-3 days of proof verification using 128 CPU cores. Since most of our evaluated models are 7B in size, the overall computational cost, while non-trivial, remains acceptable. However, those wishing to experiment with larger models or increased sampling budgets should be prepared for significantly higher compute requirements.

## M LIMITATIONS

Although our human-in-the-loop pipeline significantly enhances the robustness of FormalMATH, several limitations remain. First, there is no machine-verifiable meta-review mechanism grounded in formal reasoning Pierce et al. (2025) to rigorously ensure logical alignment between informal mathematical statements and their corresponding Lean4 formalizations. While we mitigate this by involving multiple IMO-level experts for cross-validation, the process ultimately depends on human intuition and domain knowledge rather than a fully formalized verification system, leaving the potential for subtle semantic misalignments. Second, the evaluation of FormalMATH, and, more broadly, any large-scale Lean4 benchmark, requires substantial computational resources, verifying thousands of formal proofs remains computationally intensive and time-consuming. Finally, due

to resource constraints and concurrent development timelines, we have not yet evaluated recently released models such as DeepSeek-Prover-V2 Ren et al. (2025). While we do not expect this to affect the primary contributions and novelty of our benchmark, we plan to include updated results in future versions once the results are feasible.

## N    PROMPT FOR NEGATION-BASED DISPROOF

---

**Example 3.2: Negation-Based Disproof Protocol to Filter out Non-provable Statements**

**Original Lean4 Statement:**

```
import Mathlib

def refBase (n : ℕ) : Prop :=
 ∀ k l, 0 < k → 0 < l → k < n → l < n →
 (k | n → l | n → (2 * k - l | n ∨ 2 * l - k | n))

theorem olymid_ref_base_1120 : {n | 1 < n ∧ refBase n} = {6, 9, 15} := by
 sorry
```

**Negation-based Disproof by Contradiction Construction:**

```
theorem olymid_ref_base_1120_negative : {n | 1 < n ∧ refBase n} ≠ {6, 9, 15} := by
 simp (config := { decide := true })[refBase]
 simp only [Set.ext_iff, Set.mem_setOf_eq, Set.mem_insert_iff, Set.mem_singleton_iff]
 intro h
 have h1 := h 7
 simp (config := { decide := true }) at h1
 obtain ⟨k, hk0, l, hl0, hk, hl, hkd, hld, h1, h2⟩ := h1
 interval_cases k <;> interval_cases l <;> simp_all (config := {decide := true})
```

---

## O    EXPERIMENTAL SETUPS

### O.1    MAIN EXPERIMENT

This section illustrates the experimental settings for Section 5.1.

**LLM-based Prover Settings.**

We focus on the following two different proof-generation approaches:

- **Best-First Tree-Search (BFS) Methods.** Each node in the search tree represents an intermediate proof state, and a heuristic scoring function assigns a priority to each node. We evaluate three baseline models under this category: BFS-Prover Xin et al. (2025), DeepSeek-Prover-V1.5-RL Xin et al. (2024), and InternLM-V2.5-Prover Wu et al. (2024).
- **Single-Pass Generation Methods.** The models under this category generate a complete proof in one pass, without iterative refinement or explicit intermediate states. In our paper, we consider the following baseline models: STP Dong & Ma (2025), DeepSeek-Prover-V1.5-SFT Xin et al. (2024), DeepSeek-Prover-V1.5-RL Xin et al. (2024), Goedel-Prover Lin et al. (2025), and Kimina-Prover-7B Wang et al. (2025).

**Metrics.** We evaluate theorem provers using the Pass@$K$ metric, which measures the fraction of problems for which at least one valid proof is found among the top $K$ generated attempts. (1) For BFS, $K = N \times S \times T$, where $N$ denotes the number of best-first search attempts, $S$ is the number of tactics proposed during each expansion, and $T$ is the total number of expansion iterations. (2) For SPG, $K$ corresponds to the total number of complete proof trajectories sampled from the model.

**Prompts.** In the experiments, we only consider vanilla generation strategies (see Example I.1), where models directly generate Lean4 proof without explicit requirement of chain-of-thought (CoT) rationales (natural language thoughts interleaved with Lean4) or augmenting with natural language solutions.

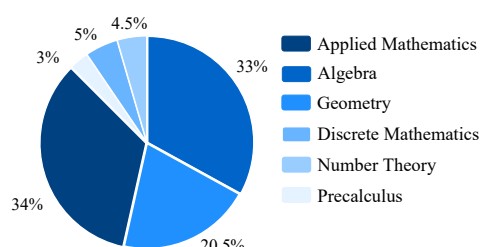
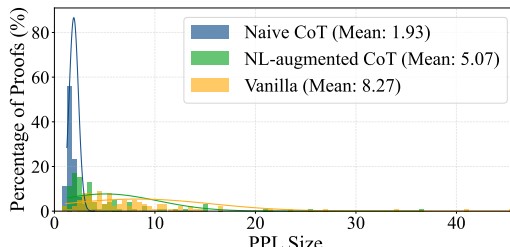

(a) Training Domains of Goedel-Prover    (b) Perplexity of DeepSeek-V1.5-SFT

Figure 6: (a) The mathematical domain distribution of Goedel-Prover's training dataset. (b) The perplexity distribution of Deepseek-V1.5-SFT across various proof generation modes.

**Verifier.** In Lean4, the correctness of proofs is verified by the compiler Leanprover Community (2023). However, verifying individual proofs is often time-consuming, largely due to the significant overhead associated with importing the Mathlib4 library Mathlib Community (2020). To mitigate this inefficiency, we use a tree-structured parallelism approach (see

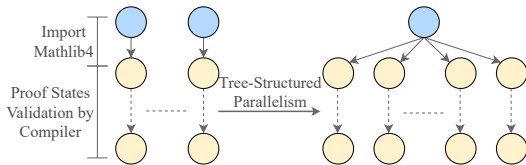

Figure 5: Our efficient Lean4 verifier implementation.

Figure 5). In this implementation, a parent thread manages the root node, which handles the computationally intensive import operations of Mathlib4. Concurrently, child threads process subsequent nodes in parallel, each corresponding to an individual proof. By centralizing the costly import operation at the root, redundant overhead is eliminated, and resources are efficiently allocated to parallelize proof verification. This simple trick effectively optimizes test-time efficiency by avoiding repeated computational overhead, ensuring scalable and efficient utilization of computational resources.

## O.2    EXPERIMENTAL SETUPS FOR EVALUATING TEST-TIME SCALING OF THEOREM PROVERS ON FORMALMATH-LITE

This section illustrates the experimental settings for Section 5.2.

Inspired by the recent success of test-time compute scaling Snell et al. (2024); Xiao et al. (2024); Muennighoff et al. (2025); Yu et al. (2025), this section examines its impact on the formal mathematical reasoning capabilities of LLM-based theorem provers using our FormalMATH benchmark. To simplify, we only evaluate BFS and repeated sampling here. To enable a systematic evaluation, we introduce FormalMATH-Lite, which is a curated subset of FormalMATH designed for efficient yet rigorous test-time scaling analysis. We compare state-of-the-art provers' performance on FormalMATH-Lite under varying sampling budgets, as shown in Table 6.

**FormalMATH-Lite.** Evaluating the full FormalMATH benchmark under large sampling budgets (*e.g.*, Pass@3200) requires prohibitively high computational resources. To enable scalable yet rigorous analysis, we propose FormalMATH-Lite, a carefully selected subset of 425 problems (comprising 359 high school-level and 66 undergraduate-level problems) designed with two critical features: (1) We utilize DeepSeek-V1.5-RL for outcome-driven difficulty assessment, evenly sampling solvable and unsolvable problems via constrained sampling budgets (*e.g.*, Pass@32). This balanced approach effectively highlights measurable scaling effects during test-time evaluation. (2) Domain Distribution Alignment: This subset follows a mathematical domain distribution similar to the full FormalMATH benchmark (algebra, calculus, discrete mathematics, etc) using stratified sampling, ensuring sufficient coverage of core disciplines. In Appendix E, we also provide the detailed distribution of FormalMATH-Lite.

**Experimental Settings.** In this experiment, we maintain identical experimental configurations to those in Section 5.1, with one critical exception: the sampling budget scales. Section 5.1 used constrained sampling budgets (*e.g.*, Pass@32) due to computational resource limitations of the full FormalMATH benchmark. Here, leveraging FormalMATH-Lite, we deploy expanded sampling budgets (*e.g.*, up to Pass@3200 for SPG and Pass@32×32×100 for BFS).

## O.3 Experimental Setups for Evaluating how CoT Enhances Model Capabilities on Formal Mathematical Reasoning

In this section, we evaluate three different reasoning strategies in Lean4 proof generations: (1) naive CoT prompting (see Example I.2), (2) NL-augmented CoT (see Example I.3): CoT augmented with natural language solution example, and (3) vanilla generation strategies (see Example I.1) via test-time scaling on FormalMATH-Lite (See Figure 3). Our goal is to *measure whether—and to what extent—informal mathematical reasoning contributes to the rigor and effectiveness of subsequently derived formal proofs.*

**Experimental Setups.** We evaluate DeepSeek-Prover-V1.5-SFT and DeepSeek-Prover-V1.5-RL (which are the only models explicitly trained with all three prompting strategies) on the FormalMATH-Lite benchmark by applying test-time scaling (up to Pass@3200).

---

**Example 4.3: naive CoT outperforms NL-augmented CoT and vanilla strategies in formal theorem proving.**

**Naive CoT (Success):**

```
theorem algebra_68653_cot : ¬ ∃ x y : ℤ, x^3 + y^4 = 7 := by
  -- Assume there are integers x and y such that x^3 + y^4 = 7.
  intro h
  -- Extract the integers x and y from the assumption.
  rcases h with ⟨x, y, hxy⟩
  -- Consider the equation modulo 13.
  have hxy_mod : (x^3 + y^4) % 13 = 7 % 13 := by rw [hxy]
  -- Calculate the possible values of x^3 modulo 13.
  have x3_mod : x^3 % 13 = 0 ∨ x^3 % 13 = 1 ∨ x^3 % 13 = 5 ∨ x^3 % 13 = 8 ∨ x^3 % 13 = 12 := by
   have : x % 13 = 0 ∨ x % 13 = 1 ∨ x % 13 = 2 ∨ x % 13 = 3 ∨ x % 13 = 4 ∨ x % 13 = 5 ∨ x % 13 = 6 ∨ x % 13 = 7 ∨ x %
       13 = 8 ∨ x % 13 = 9 ∨ x % 13 = 10 ∨ x % 13 = 11 ∨ x % 13 = 12 := by omega
   rcases this with (h | h | h | h | h | h | h | h | h | h | h | h | h) <;> simp [h, pow_succ, Int.mul_emod]
  -- Calculate the possible values of y^4 modulo 13.
  have y4_mod : y^4 % 13 = 0 ∨ y^4 % 13 = 1 ∨ y^4 % 13 = 3 ∨ y^4 % 13 = 9 := by
   have : y % 13 = 0 ∨ y % 13 = 1 ∨ y % 13 = 2 ∨ y % 13 = 3 ∨ y % 13 = 4 ∨ y % 13 = 5 ∨ y % 13 = 6 ∨ y % 13 = 7 ∨ y %
       13 = 8 ∨ y % 13 = 9 ∨ y % 13 = 10 ∨ y % 13 = 11 ∨ y % 13 = 12 := by omega
   rcases this with (h | h | h | h | h | h | h | h | h | h | h | h | h) <;> simp [h, pow_succ, Int.mul_emod]
  -- Check that x^3 + y^4 cannot be congruent to 7 modulo 13.
  rcases x3_mod with (h1 | h1 | h1 | h1 | h1) <;> rcases y4_mod with (h2 | h2 | h2 | h2) <;> simp [h1, h2,
      Int.add_emod] at hxy_mod <;> omega
```

**NL-agumented CoT (Not Success):**

```
theorem algebra_68653_nat : ¬ ∃ x y : ℤ, x^3 + y^4 = 7 := by  /–
  We aim to show that there do not exist integers \( x \) and \( y \) such that \( x^3 + y^4 = 7 \). Suppose, for the sake of
      contradiction, that such integers exist. We would then have \( x^3 + y^4 = 7 \). We analyze the possible values of \( x^3
      \mod 7 \) and \( y^4 \mod 7 \) to find a contradiction.
  1. **Lemma:** For any integer \( x \), \( x^3 \mod 7 \) can only be 0, 1, or −1.
   – Proof: By examining all possible values of \( x \mod 7 \), we find that \( x^3 \mod 7 \) is either 0, 1, or −1.
  2. **Lemma:** For any integer \( y \), \( y^4 \mod 7 \) can only be 0, 1, or 2.
   – Proof: By examining all possible values of \( y \mod 7 \), we find that \( y^4 \mod 7 \) is either 0, 1, or 2.
  3. **Combining the Lemmas:** If \( x^3 + y^4 = 7 \), then \( x^3 \mod 7 \) and \( y^4 \mod 7 \) must sum to 7 modulo 7.
      However, the possible values of \( x^3 \mod 7 \) are 0, 1, or −1, and the possible values of \( y^4 \mod 7 \) are 0, 1, or
      2. None of these combinations can sum to 7 modulo 7, leading to a contradiction.
  Thus, we conclude that there do not exist integers \( x \) and \( y \) such that \( x^3 + y^4 = 7 \).
  –/
  -- Assume for contradiction that there exist integers x and y such that x^3 + y^4 = 7.
  intro h
  obtain ⟨x, y, hxy⟩ := h
  -- Analyze the possible values of x^3 modulo 7.
  have hx : x^3 % 7 = 0 ∨ x^3 % 7 = 1 ∨ x^3 % 7 = 6 := by
   have hx : x % 7 = 0 ∨ x % 7 = 1 ∨ x % 7 = 2 ∨ x % 7 = 3 ∨ x % 7 = 4 ∨ x % 7 = 5 ∨ x % 7 = 6 := by
     omega
   rcases hx with (h | h | h | h | h | h | h) <;>
     simp [h, pow_three, Int.mul_emod, Int.add_emod]
  -- Analyze the possible values of y^4 modulo 7.
  have hy : y^4 % 7 = 0 ∨ y^4 % 7 = 1 ∨ y^4 % 7 = 2 := by
   have hy : y % 7 = 0 ∨ y % 7 = 1 ∨ y % 7 = 2 ∨ y % 7 = 3 ∨ y % 7 = 4 ∨ y % 7 = 5 ∨ y % 7 = 6 := by
     omega
   rcases hy with (h | h | h | h | h | h | h) <;>
     simp [h, pow_four, Int.mul_emod, Int.add_emod]
  -- Combine the results to find a contradiction.
  have h7 : (x^3 + y^4) % 7 ≠ 7 := by
   rcases hx with (h | h | h) <;>
     rcases hy with (h' | h' | h') <;>
       simp [h, h', Int.add_emod, Int.mul_emod]
```

---

```
-- Conclude the proof by contradiction.
omega
```

**Vanilla (Not Success):**

```
theorem algebra_68653_vanilla : ¬ ∃ x y : ℤ, x^3 + y^4 = 7 := by
  intro h; rcases h with ⟨x, y, hxy⟩; have hx : x^3 ≥ 0 := by positivity; have hy : y^4 ≥ 0 := by positivity
linarith
```

### O.4    EXPERIMENTAL SETUPS FOR DELVING INTO COMMON ERROR PATTERNS OF EXISTING PROVERS

In this section, we systematically analyze common error patterns observed in existing theorem provers (*e.g.*, DeepSeek-V1.5, STP, Goedel, and Kima-Prover). We employ advanced general-purpose LLMs (*e.g.*, o4-mini) to automate both error diagnosis and classification: first extracting salient verbalized features (*i.e.*, Example J), then assigning the found error features via a second prompt-driven call (*i.e.*, Example K). For each prover, we randomly sampled 100 failed proofs from a variety of Lean4 statements and processed them through our two-stage diagnosis and classification pipeline. Human domain experts then manually reviewed and corrected both the extracted features and the preliminary labels. We identified the four most common failure patterns—incomplete proofs, inability to handle complex inequalities, improper use of automation tactics, and redundant hypothesis introduction—as summarized in Table 8. Note that a single proof attempt may exhibit multiple errors, so the percentages do not sum to 100%.

## THE USE OF LARGE LANGUAGE MODELS (LLMs)

In this work, Large Language Models (LLMs) were exclusively utilized for grammar correction and text polishing during the manuscript preparation stage. No aspects of the research methodology, ideation, experiment design, data processing, result analysis, or core algorithm implementations involved LLMs in any capacity. All scientific contributions, originality, and rigor are ensured by the authors. As per ICLR policy, we confirm that LLMs did not play an important, original, or non-standard role in the development of this research.

| **Lean4 Statement** | **Prover Output** | **Explanation** |
|---|---|---|
| `theorem` omni_theorem_4000 (f $: \mathbb{R} \to \mathbb{R}$) : ($\forall$ x y, (x + y ^ 2) * f (y * f x) = x * y * f (y ^ 2 + f x)) $\leftrightarrow$ $\forall$ x, f x = x $\lor$ f x = 0 $\lor$ $\exists$ a, a $\leq$ −1 $\lor$ a > 0 $\land$ f x = if x $\neq$ −a ^ 2 then 0 else a := `by` | -- Proof Before
-- If f(x) $\neq$ 0, we need to show that there exists an a such that the conditions hold.
right
right
`use 0`
-- We choose a = 0 to satisfy the conditions.
`simp`
--Proof After | **Improper use of Automation Tactics.** The prover selects the witness $a = 0$ without checking the original side-condition and then calls `simp` to close off the goal. This bypasses the essential case-by-case analysis of the core functional equation, resulting in an invalid proof that never constructs the required nonzero witness when needed. |
| `theorem` DEMIMathAnalysis_50 : Tendsto ($\lambda$ n => sqrt n * $\int$ x in univ, 1 / (1 + x ^ 2) ^ n) atTop (N (sqrt $\pi$)) := `by` | rw [**show** sqrt = **fun** x => sqrt x **by** funext; rfl]
`apply tendsto_atTop_mono` (**fun** n => sqrt_le_sqrt <\| integral_mono_on _ _ <\| **fun** x _ => **by** gcongr all_goals norm_num)
refine' tendsto_atTop_of_eventually_const (**by** simp) all_goals filter_upwards **with** _
`;¿ simp` | **Incomplete Proof.** The proof invokes tendsto_atTop_mono without establishing any of the analytical prerequisites. The final `simp` steps trivially handle eventual constancy but leave the main asymptotic argument unresolved. |
| `theorem` aime_all_2005_II_1 (n : $\mathbb{N}$) (hn : n $\geq$ 6) (h : {s : Finset (Fin n) \| s.card = 6}.ncard = 6 * {s : Finset (Fin n) \| s.card = 3}.ncard) : n = 13 := `by` | norm_num [Nat.choose] **at** h
revert n hn h
`rintro n hn h`
ring_nf **at** h
nlinarith | **Redundant Hypothesis Introduction.** After norm_num already rewrites the binomial coefficients, the revert ; rintro sequence merely reintroduces n, hn, and h in the same form, adding no new information and cluttering the context. |
| `theorem` algebra_528739 (a b c : $\mathbb{R}$) (ha : 0 < a) (hb : 0 < b) (hc : 0 < c) : 1 / (a^3 + b^3 + a * b * c) + 1 / (b^3 + c^3 + a * b * c) + 1 / (c^3 + a^3 + a * b * c) $\leq$ 1 / (a * b * c) := `by` | -- We start by simplifying the left-hand side using the fact that each term is non-negative.
**have** $h_0$ : 0 < a * b * c := **by** positivity
...
-- We then apply the inequality to each term.
rw [div_le_div_iff] <;>
`nlinarith`, [sq_nonneg (a − b), sq_nonneg (b − c), sq_nonneg (c − a)]
... | **Inadequate Handling of Inequalities.** The solver attempts to apply nlinarith after a single div_le_div_iff, but the cyclic, high-degree fractional structure exceeds its linear-and-quadratic reasoning scope. |

Table 11: Examples of common Lean4 error patterns in LLM-based provers.

