# OpenReview forum: "FormalMATH: Benchmarking Formal Mathematical Reasoning of Large Language Models"
_ICLR.cc/2026/Conference — ICLR 2026 Conference Desk Rejected Submission_

### Official Review · Reviewer_fQki · 2025-10-28

**Soundness:** 3
**Presentation:** 2
**Contribution:** 2
**Rating:** 2
**Confidence:** 4

**Summary:**

This paper introduces a new benchmark to evaluate the performance of large language models on formal mathematical proving across various topics at the high school and undergraduate levels. It situates the work in relation to existing datasets, emphasizing the benchmark’s larger scale and broader topic coverage. The authors conduct empirical evaluations to analyze the weaknesses of current LLM-based provers and summarize several common error patterns. To ensure correctness and formal validity, the benchmark is constructed through a human-in-the-loop process involving both LLMs and human experts. The syntax of the generated problems is verified using Lean 4, the semantics are checked by other LLMs, and final validation is performed by human reviewers.

**Strengths:**

●	The authors have made significant efforts to collect and compile a large set of mathematical proof problems for the proposed benchmark. The resulting dataset covers a much broader range of topics and difficulty levels than existing benchmarks, substantially extending the scope of the math-proving domain.

●	The increased scale of the benchmark offers opportunities for further improving, fine-tuning, or post-processing LLMs for formal reasoning tasks. It also provides a convenient and comprehensive resource for researchers and practitioners interested in evaluating models across diverse mathematical areas and levels of difficulty.

**Weaknesses:**

●	While I appreciate the authors’ effort to present detailed findings and observations from using the proposed benchmark to evaluate existing LLMs, very little information about the dataset itself is provided in the main paper. This limits the paper’s clarity and informativeness. It would be helpful if the authors included a brief discussion of the benchmark’s data format, data collection process, and implemented evaluation metrics. If space is a concern, the presentation of Table 2 could be optimized to make room for these essential details.

●	Continuing from the last comment on information clarity, the description of the evaluation setup is missing in Sections 4 and 5, which makes the paper less self-contained without the appendix. A concise summary of the experimental configuration would improve readability and reproducibility.

●	The paper points out that existing formal mathematics benchmarks are limited in scope, focusing mainly on a few high school or college-level topics. While I appreciate the broader coverage this benchmark brings, the paper should give stronger justification for the kind of depth and qualitative diversity it adds that’s beyond simply combining more problems.

●	The benchmark mainly reports the Pass@K metric, which offers a limited view of performance. Adding other evaluation measures such as final proof accuracy, intermediate reasoning correctness, or step-by-step reasoning quality would give a more complete picture of model performance.

**Questions:**

●	In Figure 2, both the undergraduate and high school math domains include Precalculus. Are there any overlapping questions between these two categories? If both domains involve Precalculus, how was it decided which problems belong to each level?

●	What other metrics might be useful for evaluating LLMs’ mathematical proving abilities beyond Pass@K? If such metrics exist, how difficult would it be to incorporate them into the current evaluation framework?

●	Could the authors provide more details on the choice of the Lean 4 framework and what this decision implies in practice? How challenging is it to represent or formalize a mathematical proving problem within the Lean 4 environment?

---

> ### Author Response · Authors · 2025-11-22
> **Responses to Reviewer fQki (Part 1/3)**
>
> We sincerely thank the reviewer for their constructive and thorough comments on our work. We take every comment seriously and hope our response addresses the reviewer's concerns. If any questions remain, we are happy to address them.
>
> ---
>
> ### **1. W1: included a brief discussion of the benchmark’s data format, data collection process, and implemented evaluation metrics in the main paper.**
>
> Thank you for your great suggestions! We have included the details about data format (Section 3.2), collection process (Section 3.3), and the implemented evaluation metrics (Section 4.1) in the paper.
>
> ### **2. W2: add evaluation setup in Sections 4 and 5**
>
> Good idea! We have reformat the Table reviewer mentioned, and added the evaluation setup in Sections 4 and 5 in the main paper.
>
>
> ### **3. W3: stronger justification for depth and qualitative diversity beyond simply combining more problems**
>
> We thank the reviewer for the suggestions. We compare the diversity and depth of FormalMATH with existing benchmarks. We compare against: 1. MiniF2F, 2. Leanworkbook, 3. Putnam.
>
> Experimental Setup: We use GPT-4o-mini (temperature=0.7, max tokens=4k) as an LLM-based judge to compare difficulty, depth, and diversity across benchmarks. For each comparison, using three different random seeds (37, 42, 50), we randomly sample 20 Lean4 statements from FormalMATH and 20 statements from the counterpart benchmark each time. The two sets of 20 statements are then fed into the LLM to perform comparison. The same procedure is repeated 30 times in total. The final result is averaged across these 30 times comparison.
>
> We ask the LLM to score both sides on: (1) diversity (topic diversity, conceptual diversity, and structural diversity, including different proof structures and theorem types), (2) difficulty, and (3) depth.
>
> **Prompt for evaluating diversity**:
>
> ```jsx
> You are evaluating the diversity of a set of formal mathematical statements written in Lean4.
>
> Here are {len(statements)} statements:
>
> {statements_text}
>
> Please evaluate the DIVERSITY of these statements on a scale of 0.0 to 1.0, where:
> - 0.0 = Very low diversity (statements are very similar, cover similar topics/concepts)
> - 0.5 = Moderate diversity (some variation in topics, concepts, or approaches)
> - 1.0 = Very high diversity (statements cover diverse topics, use different concepts, approaches, and mathematical domains)
>
> Consider:
> - Topic diversity (different mathematical areas: algebra, geometry, analysis, number theory, etc.)
> - Conceptual diversity (different types of mathematical concepts used)
> - Structural diversity (different proof structures, theorem types)
> - Notation diversity (variety in mathematical notation and formalization styles)
>
> Respond with ONLY a single number between 0.0 and 1.0 (e.g., 0.75), nothing else.
> ```
>
> **Prompt for evaluating depth**:
>
> ```jsx
> You are evaluating the depth of a set of formal mathematical statements written in Lean4.
>
> Here are {len(statements)} statements:
>
> {statements_text}
>
> Evaluate the DEPTH of the statements, consider:
> - theoretical depth and significance
> - whether statements reveal deep mathematical insights
>
> Give a score between 0.0 and 1.0, where 0.0=shallow and 1.0=very deep.
>
> ```
>
> **Prompt for evaluating difficulty:**
>
> ```jsx
> You are evaluating the difficulty of a set of formal mathematical statements written in Lean4.
>
> Here are {len(statements)} statements:
>
> {statements_text}
>
> Evaluate the DIFFICULTY of the statements, consider:
> - the required level of mathematical knowledge
> - the complexity of the concepts
> - the difficulty of the proofs
>
> Give a score between 0.0 and 1.0, where 0.0=very easy and 1.0=very difficult.
> ```
>
> And we got the following results
>
> |  | **Lean-workbook** | **MiniF2F** | **Putnam** | **FormalMATH-All** |
> | --- | --- | --- | --- | --- |
> | **Diversity**  | 0.570 ± 0.060 | 0.688 ± 0.046 | 0.608 ± 0.076 | **0.757 ± 0.031** |
> | **Difficulty** | 0.560 ± 0.060 | 0.455 ± 0.098 | **0.778 ±0.044** | 0.768 ±0.067 |
> | **Depth** | 0.645 ± 0.033 | 0.652 ± 0.058 | 0.720 ± 0.048 | **0.765 ± 0.045** |
>
> Key observations:
>
> 1. Highest Diversity while maintaining difficulty: FormalMATH scores highest on diversity while matching the hardest benchmarks (Putnam) in difficulty.
> 2. Depth advantage: It outperforms Putnam on depth (+0.045) despite containing 10× more statements, showing that scale was achieved without loss of sophistication.

---

> ### Author Response · Authors · 2025-11-22
> **Responses to Reviewer fQki (Part 2/3)**
>
> ### **4. W4: adding other evaluation measures such as final proof accuracy, intermediate reasoning correctness, or step-by-step reasoning quality**
>
> Thank you for this important suggestion regarding evaluation measures. We realize we need to clarify why **Pass@K** is the standard and most meaningful metric for formal mathematical proving.
>
> 1. Why we use Pass@K
>
> **Pass@K** measures whether a model can generate **at least one** formally verified proof within K attempts. In formal math reasoning, the evaluation is GT-free, which means that we don’t need n existing answers to validate the correctness of proof, instead, we need proof assistant, which use formal logic to verify the correctness of proof. However, due to the inherent difficulty of formal reasoning, we allow provers to attempt several times. In existing literatures (e.g. Lean4 proof completion paper [2][3][4][5]), Pass@K is the most common metric used in LLM.
>
> 1. Actions taken to follow the reviewer’s suggestions
>
> In Lean4 benchmarks (e.g., MiniF2F, ProofNet, and FormalMATH), ground truth answers are not provided because: (1) all Lean4 benchmarks focus on proof problems rather than calculation problems, and (2) correctness is determined by the Lean4 proof assistant. Thus, we have no reference answer to compare against.
>
> To address the reviewer's concerns, we measure 1. final proof accuracy using Pass@1 correctness, where correctness is determined by the proof assistant (Lean4 compiler). Note that in proof completion scenarios, when the compiler verifies a proof as correct, it is 100% correct in both semantics and syntax. Also, we use GPT-4o-mini (temperature=0.7, max tokens=4k) as an LLM-based judge to compute the step-by-step reasoning quality and constrain it to output a 0.0-1.0 score as an indicator of intermediate reason quality.
>
> ```jsx
> You are evaluating the **step-by-step reasoning quality** of a Lean 4 proof fragment.
>
> **Input**
> --------
> Theorem statement:
> {theorem_statement}
>
> Proof candidate (Lean 4):
> {proof_code}
>
> **Task**
> --------
> 1. Read the tactic sequence line-by-line.
> 2. Rate **how well the proof explains its own logic**:
>    - Are intermediate goals clearly justified?
>    - Are tactics chosen appropriately for the goal they face?
>    - Is there a coherent high-level plan (e.g., inequality ⟹ rewrite ⟹ linear arithmetic)?
> 3. Assign **one** score 0.0 – 1.0:
>    - 0.0 = no discernible reasoning, random tactics
>    - 0.5 = partial or weak justifications, large gaps
>    - 1.0 = clear, step-by-step reasoning with appropriate tactics
>
> **Output format (strict)**
> -------------------------
> Reasoning: <one sentence describing the strongest weakness or strength>
> Score: <a single float with exactly two decimals, e.g., 0.63>
> ```
>
> We thus have the following result evaluate on FormalMATH-All:
>
> |  | Accuracy  | Step-by-Step Reason Quality |
> | --- | --- | --- |
> | DeepSeek-Prover-V1.5-RL | 6.21% | 0.65 |
> | Goedel-V1 | 8.72% | 0.71 |
> | STP | 9.27% | 0.71 |
> | DeepSeek-V2 | 11.95% | 0.74 |
> | Goedel-V2 | **13.20%** | **0.75** |
>
> We observe that the step-by-step reason quality aligns with the final accuracy of the provers, and well reflect the performance of provers.
>
> We thus thank the reviewer for providing these great suggestions regarding the metrics.
>
> ### **5. Q1: both the undergraduate and high school math domains include Precalculus. Are there any overlapping questions between these two categories? If both domains involve Precalculus, how was it decided which problems belong to each level?**
>
> Thank you for your question. In Appendix G, we present the prompt used to classify Lean4 math problems via LLM. Our taxonomy follows the same structure as OmniMATH for the classification tags you mentioned.
>
> We maintain a clear standard to distinguish precalculus (limits, trigonometric functions, etc.) from calculus (primarily derivatives and integrals). Classification is determined by the concrete problem type rather than by whether it appears in a college or high school curriculum. For problems sourced from OmniMATH, we directly use the original labels. U-math, HardMath, and DEMIMATH all provide ground truth labels.
>
> We provide the following examples to show precalculus and calculus problems in our benchmark.
>
> **Precalculus (Limit)**:
>
> ```jsx
> import Mathlib
> open Topology Filter Real Complex TopologicalSpace Finset
> open scoped BigOperators
>
> theorem DEMIMathAnalysis_2
>   (a b : ℕ → ℝ)
>   (α β : ℝ)
>   (h₁ : Tendsto a atTop (𝓝 α))
>   (h₂ : Tendsto b atTop (𝓝 β)) :
>   Tendsto (λ n => (∑ i in range (n + 1), a i * b (n - i)) / n) atTop (𝓝 (α * β)) := by
> ```
>
> **Calculus (Integral)**:
>
> ```jsx
> import Mathlib
>
> open Real
>
> theorem u_math_812 : ∫ u in (exp 1)..(exp 2), ∫ v in (log u)..2, (v + log u) = exp 2 - exp 1 / 2 := by
> ```

---

> ### Author Response · Authors · 2025-11-22
> **Responses to Reviewer fQki (Part 3/3)**
>
> ### **6. Q2: What other metrics might be useful for evaluating LLMs’ mathematical proving abilities beyond Pass@K? If such metrics exist, how difficult would it be to incorporate them into the current evaluation framework?**
>
> We thank the reviewer for suggesting evaluation metrics beyond Pass@K, e.g. the additional accuracy metric and step-by-step reasoning quality. We implemented these metrics on FormalMATH and tested a few state-of-the-art provers. Results are shown in W4.
>
> ### **7. Q3: Could the authors provide more details on the choice of the Lean 4 framework and what this decision implies in practice? How challenging is it to represent or formalize a mathematical proving problem within the Lean 4 environment?**
>
> Thank you for this important question. We realize our original manuscript didn't sufficiently elaborate on the rationale behind Lean 4 and the practical challenges of formalization. Here's a detailed explanation and we will incorporate it into the revised manuscript.
>
> **Why Lean4**
>
> First,  Unlike natural language mathematics, where correctness is judged by a final answer marked in a \box{}, formal mathematics in Lean 4 determines correctness via a proof assistant grounded in dependent type theory. This means our benchmark does not require manually verified ground-truth answers, the proof assistant automatically and rigorously checks correctness for us. Thus we choose formal language over natural language.
>
> Second, we select lean4 over other alternatives (Coq, Isabelle/HOL) based on several practical considerations:
>
> 1. **Open-soruce library support**: The most comprehensive active mathematics library with >1M lines of code covering diverse domains in our benchmark (calculus, algebra, number theory, discrete math). This prevents us from having to formalize basic definitions from scratch.
> 2. **Open-soruce model support:** Unlike other proof assistants, Lean 4 has a growing ecosystem of open-source LLM-based provers (e.g., DeepSeek, Goedel, Kimina). This enables direct comparison and reproducibility, which is critical for benchmarking. In contrast, state-of-the-art Isabelle provers are often closed-source or rely on proprietary frameworks, limiting community-driven progress.
>
> **How challenging is formalization in Lean4**
>
> Formalization with Lean4 is extremely challenging even for humans. For example, the Polynomial Freiman–Ruzsa conjecture [6] has existing proofs, yet Terence Tao and his collaborators have spent several years in formalizing these proofs manually, a project that is still ongoing. This demonstrates that even with existing natural language proofs, formalizing them in Lean4 remains highly challenging.
>
> ### Reference
>
> [1] https://arxiv.org/pdf/1701.03322
>
> [2] DeepSeek-Prover-V1.5: Harnessing Proof Assistant Feedback for Reinforcement Learning and Monte-Carlo Tree Search https://arxiv.org/pdf/2408.08152v1
>
> [3] Goedel-Prover-V2: Scaling Formal Theorem Proving with Scaffolded Data Synthesis and Self-Correction ****https://arxiv.org/pdf/2508.03613
>
> [4] STP: Self-play LLM Theorem Provers with Iterative Conjecturing and Proving https://arxiv.org/pdf/2502.00212
>
> [5] BFS-Prover: Scalable Best-First Tree Search for LLM-based Automatic
> Theorem Proving https://arxiv.org/pdf/2502.03438
>
> [6] https://terrytao.wordpress.com/tag/polynomial-freiman-ruzsa-conjecture/

---

> ### Author Response · Authors · 2025-11-27
>
> Dear Reviewer fQki,
>
> We sincerely thank you once again for your time and valuable feedback on our manuscript. We hope that the clarifications provided, along with the new experiments and discussions included in the revised version, have fully addressed your concerns.
> Should you find the revisions satisfactory, we would greatly appreciate it if you could consider updating your score to reflect the additional results and improvements.
> We remain fully available to address any further comments or questions you may have during the discussion phase.
>
> Thank you again for your thoughtful and constructive review.
>
> Best regards,
>
> The authors

---

### Official Review · Reviewer_edMV · 2025-10-29

**Soundness:** 2
**Presentation:** 3
**Contribution:** 3
**Rating:** 6
**Confidence:** 4

**Summary:**

This paper introduces "FormalMATH," a new, large-scale, and challenging benchmark for formal theorem proving, spanning from Olympic style problems to undergraduate mathematics. Its preparation involved auto-formalization by a specialized model, semantic verification via LLMs, and final expert validation, resulting in 5,560 problems.

A comprehensive evaluation of current models on this benchmark reveals several key findings. First, overall performance is poor. Second, there are significant performance variations among models across different domains. Third, test-time scaling offers only subtle improvements. Finally, while the default Chain-of-Thought (CoT) prompting is beneficial, requiring a CoT plan before the formal proof is counterproductive.

The paper also attributes the models' failures to several common issues: resorting to inappropriate tactics for unsolvable problems, producing incomplete proofs with placeholders, struggling with complex inequalities, and generating redundant hypotheses.

**Strengths:**

The paper presents a highly automated pipeline for benchmark creation, which has successfully produced a large-scale dataset. This dataset is notable for its breadth, spanning a wide range of mathematical domains, and its depth, covering difficulty levels from undergraduate math down to the Olympiad level. This high degree of automation suggests strong potential for future scalability.

The evaluation of model performance is multi-faceted and meticulous. The analysis across different mathematical domains, the investigation of test-time scaling, and the study of Chain-of-Thought (CoT) prompting provide valuable observations.

The error analysis is very detailed. The identified issues (e.g., resorting to inappropriate tactics, producing incomplete proofs, struggling with inequalities, generating redundant hypotheses) are specific and actionable, providing direct insights that can inform engineering efforts to optimize LLM-based provers.

**Weaknesses:**

Regarding the Analysis and Pipeline:
- Performance Decline on Advanced Problems: The accuracy of the proposed pipeline shows a notable decline when applied to more advanced mathematical areas. Specifically, the success rate for the competition-level AIME problems was 37.6% (371/934), whereas it dropped to just 4.6% (67/1466) for the undergraduate-level HardMath problems. This indicates potential obstacles in formalizing high-difficulty problems within the pipeline.
- Benchmark Imbalance and Potential for Overestimation: The imbalance between Olympiad-level and undergraduate-level problems in the final benchmark may lead to an overestimation of a model's true capabilities. It is plausible that the types of problems a model is good at formalizing are also the types of problems it is good at solving, creating a potential bias in the evaluation.
- Unsound Conclusions from Key Findings: The paper's explanations for some of its key findings could be strengthened.
  - The data supporting Finding 4 (the negative effect of pre-defined CoT plans) appears inconclusive. The success rates of the compared methods (51.7%, 51.2%, and 49.8% for DeepSeek-V1.5-RL) are too close to draw a strong conclusion. The narrow margin of difference (less than 2%) makes this finding seem unsound.
  - The explanation for Finding 2 (performance variation across domains) may be incomplete. The paper suggests this is due to data distribution imbalances, but this may not be the only factor. For instance, the "precalculus" domain accounts for only 3% of the training data (Figure 8a) yet achieves the second-highest success rate of 33.71% (Figure 3). This discrepancy suggests that other factors, such as the different skills tested by each domain, could also contribute to this performance variance.

Regarding Misleading Presentation:

- Lack of Context for Table 2: In Table 2, the lower part displays relative performance on the FormalMATH-lite test set. The main text does not clarify that this test set was specifically constructed from a 50/50 split of problems that DeepSeek 1.5 RL could and could not solve, which means the model implicitly serves as a baseline. This lack of clarification could cause confusion for the reader.
- Issues with Figure 3: In Figure 3, the axes are not labeled, though one can infer they represent success rates. More significantly, the scales for different domains appear inconsistent. Plotting them on a single chart in this manner allows for comparing models within a domain but can be seriously misleading when comparing performance across different domains.

Minor Issues with Specific Examples:

- Error in Example B.2: There appears to be a translation error from natural language to the formal statement. The original problem asks one to prove that two quantities approaching zero are of the same order, whereas the formalized result aims to prove that the difference between these two quantities approaches zero.
- Inappropriate Suggestion in Section 5.1: In the "Inabilities to Handle Complex Inequalities" part of subsection 5.1, the suggestion to prove a^3 + b^3 + abc ≥ abc using the AM-GM inequality seems inappropriate. The inequality simplifies directly to a^3 + b^3 ≥ 0, and its connection to AM-GM is not apparent.

If the identified issues, particularly those concerning the soundness of the findings and the clarity of the data presentation, were to be addressed, the paper would likely merit a higher score.

**Questions:**

1. Regarding the performance drop between AIME and HardMath problems, have you investigated the specific reasons or pipeline stages responsible for this decreased formalization success?
2. Could you clarify how a semantic error like the one noted in Example B.2 was able to pass through the pipeline's verification stages?

---

> ### Author Response · Authors · 2025-11-22
> **Responses to Reviewer edMV (Part 1/4)**
>
> We sincerely thank the reviewer for their constructive and thorough comments on our work. We take every comment seriously and hope our response addresses the reviewer's concerns. If any questions remain, we are happy to address them.
>
> ---
>
> ### **1. W1: The accuracy of the proposed pipeline shows a notable decline when applied to more advanced mathematical areas. Specifically, the success rate for the competition-level AIME problems was 37.6% (371/934), whereas it dropped to just 4.6% (67/1466) for the undergraduate-level HardMath problems. This indicates potential obstacles in formalizing high-difficulty problems within the pipeline.**
>
> Great question! We recognize that Table 6 may cause confusion. The reported values of 1466 and 2830 represent the size of the original HardMath and DEMIMATH datasets, not the portion we used for autoformalization in FormalMATH. We did not use all problems from these datasets, as some are not directly usable, for example, they may contain multimodal information such as images.
>
> |  | Omni-math | Numina-Olympiad | AIME-Math | BlueMO  | U-Math | Hardmath | DEMIMATH  |
> | --- | --- | --- | --- | --- | --- | --- | --- |
> | Origin | 4.4k | 11.8k | 934 | 3k | 1,100 | 1,060 (used in our pipeline) / 1,466 (all) | 88 (used in our pipeline) / 2,830 (all) |
> | Lean4 compiler | 4.3k | 10.3k | 933 | 2.9k  | 871 | 1022 | 86 |
> | LLM for semantic judge | 1.5k | 3.4k | 380 | 1,1k | 370 | 68 | 46 |
> | Negation + Human | 1.2k | 2.4k | 371 | 1,099 | 358 | 67 | 46 |
>
> However, the conservation rate for Hardmath is still subtle. The difference in conversion may be due to several reasons. For example, some formulations of natural language problems are naturally easier to formalize (e.g., proof problems versus calculation problems). Additionally, Mathlib, the repository we import for formalization, cannot cover all mathematical subjects, and the training dataset used to train the formalizer may also be imbalanced in terms of mathematical subjects.
>
> We want to note that currently there is no method that can rigorously ensure the correctness of autoformalization, all judgments are subjective. Potential methods to mitigate the low conversion rate may be to: (1) synthesize more formal-informal pairs to train better formalizers [1], or (2) train critic models that give effective suggestions for revising formalization [2]. However, all these methods cannot 100% guarantee that their formalization is correct.
>
> ### **2. W2: The imbalance between Olympiad-level and undergraduate-level problems in the final benchmark may lead to an overestimation of a model's true capabilities. It is plausible that the types of problems a model is good at formalizing are also the types of problems it is good at solving, creating a potential bias in the evaluation.**
>
> Thanks for the question! We agree with the reviewer that there could indeed be an overestimation of a model’s true capabilities. However, at the current stage, we can only rely on LLM-based autoformalization to formalize problems at scale. Otherwise, we have to fully rely on human formalization, which is known to be extremely challenging and time-consuming.
>
> However, we also want to clarify that FormalMATH focuses on formalization correctness and evaluates problems that can be formalized by current formalizers. Even if such overoptimization exists, the best-performing prover, DeepSeek-Prover-V2 (671B), can only achieve 28.31% on the benchmark. In the light of it, we can still reliably quantify and compare the capabilities of different provers using the benchmark, even if all these provers may indeed over-perform. In our paper, we focus on rigorously ensure the correctness of the formalized problems and benchmark/compare existing LLM provers. Our paper has well served such purposes.

---

> ### Author Response · Authors · 2025-11-22
> **Responses to Reviewer edMV (Part 2/4)**
>
> ### 3. W3 Part 1: The data supporting Finding 4 (the negative effect of pre-defined CoT plans) appears inconclusive. The success rates of the compared methods (51.7%, 51.2%, and 49.8% for DeepSeek-V1.5-RL) are too close to draw a strong conclusion. The narrow margin of difference (less than 2%) makes this finding seem unsound.
>
> Thank you for raising this question.
>
> In Figure 4, we compare vanilla, NL, and CoT under test-time scaling. We conclude that naive CoT outperforms NL-augmented CoT because across nearly all budgets, from Pass@1 to Pass@3200 (which is a very large sampling budget). DeepSeek-V1.5-RL and -SFT consistently score 0.5–2 points lower when initialized with human-written natural-language proofs versus self-generated CoT. This conclusion is not based solely on the result at K=3200.
>
> To further demonstrate the negative effect of NL-augmented CoT, we conducted an additional experiment on Goedel-SFT. Since it is already a CoT model, we compared two variants: Goedel-NL and Goedel-CoT across sampling budgets from Pass@1 to Pass@128.
>
> | Prover | Pass@1 | Pass@32 | Pass@128 | Mean ppl  |
> | --- | --- | --- | --- | --- |
> | **Goedel-NL** | 15.32%  | 37.30% | 41.13% | 4.06 |
> | **Goedel-CoT** | **38.12%** | **46.70%** | **48.08%** | **1.77** |
>
> We observe that the gap persists with Goedel-SFT, the CoT model constantly outperforms its -NL counterpart. Moreover, the perplexity of also shows that Goedel is more confident under CoT while less certain when augmented by NL, which aligns with our conclusion in the paper.
>
> ### **4. W3 Part 2: The explanation for Finding 2 (performance variation across domains) may be incomplete. The paper suggests this is due to data distribution imbalances, but this may not be the only factor. For instance, the "precalculus" domain accounts for only 3% of the training data (Figure 8a) yet achieves the second-highest success rate of 33.71% (Figure 3). This discrepancy suggests that other factors, such as the different skills tested by each domain, could also contribute to this performance variance.**
>
> Thank you for the suggestion. Performance imbalance could stem from multiple factors. Training data imbalance may be one, and mathematical skills (as you suggested) is another important factor. For example, Mathlib, one of the most important repositories in Lean4, has imbalanced mathematical primitives across different domains (e.g., the number of lemmas in algebra far exceeds those in geometry).
>
> ### **5. W4 Part 1: In Table 2, the lower part displays relative performance on the FormalMATH-lite test set. The main text does not clarify that this test set was specifically constructed from a 50/50 split of problems that DeepSeek 1.5 RL could and could not solve, which means the model implicitly serves as a baseline. This lack of clarification could cause confusion for the reader.**
>
> Great question! We thank the reviewer for raising this issue. The -Lite dataset selection could be biased, but we change to another perspective to observe the difficulty of FormalMATH. The performance gap is in fact a design choice. Specifically, the Lite dataset was designed based on the following principles:
>
> 1. It should be easier than the original full set so that we can observe more clearly how performance changes across different Pass@K budgets.
>
> 2. It should be small, so that the budget is affordable.
>
> 3. Overall, a smaller and moderately easier evaluation set will be beneficial for studying the effect of test-time scaling strategies.
>
> 4. The number of covering domains should be as comprehensive as the full dataset. Thus we need to make sure the Lite cover as many domains as possible.
>
> Here we use different selection policies and observe their impact on Pass@32 for Goedel-Prover-V1:
>
> | **Selection Policy** | **Pass@32** | Size | Difficulty of the Statements | # Domains |
> | --- | --- | --- | --- | --- |
> | Original -All  | 0.13  | 5,560 | 4.89 | **15** |
> | Original -Lite (Manual Difficulty Control) | 0.47 | **425** | **3.68** | 13 |
> | Single-Prover Resample (dpsk-RL-1.5) | **0.51** | 425 | 3.77 | 10 |
> | Multi-Prover Resample | 0.36 | 425 | 4.07 | 12 |
> | Random Resample | 0.17 | 425 | 4.93 | 13 |
>
> The difficulty of informal statements is determined by an LLM judge using the same difficulty classification prompt as OmniMath [3]. We use GPT-4o-mini as the judge.
>
> **Conclusion**:
>
> 1. The domain coverage of the original -Lite is closest to the full dataset.
> 2. The original -Lite dataset is a simple version of FormalMATH-All.
> 3. Random resampling makes the dataset too difficult and may not be suitable for studying test-time scaling.
>
> Thus, although the selection could be biased, the findings based on test-time scaling on FormalMATH-Lite would not change significantly. We will add these experiments on the various Lite dataset to the revised manuscript to ensure that conclusions remain consistent across different subsets.

---

> ### Author Response · Authors · 2025-11-22
> **Responses to Reviewer edMV (Part 3/4)**
>
> ### **6. W4 Part 2:  Issues with Figure 3: In Figure 3, the axes are not labeled, though one can infer they represent success rates. More significantly, the scales for different domains appear inconsistent. Plotting them on a single chart in this manner allows for comparing models within a domain but can be seriously misleading when comparing performance across different domains.**
>
> Thanks for your suggestion, we have changed the radar figure to table.
>
> ### **7. MW1: Error in Example B.2: There appears to be a translation error from natural language to the formal statement. The original problem asks one to prove that two quantities approaching zero are of the same order, whereas the formalized result aims to prove that the difference between these two quantities approaches zero.**
>
> Thank you for pointing out the discrepancy between the formalized transformation and the original text in Example B.2. We fully agree with you. The mistake may have occurred because Lean is a proof assistant that requires stronger and more rigorous conditions than the natural language version of problems.
>
> The core requirement of the original text only needs to verify the two quantities belong to the same decay order. It merely requires that the ratio of the two is bounded above and below by non-zero constants, without involving the convergence of specific errors. The formal statement represents the same order claim as a difference approaching zero, which is a stronger condition tied to asymptotic equivalence. From the perspective of logical implication, asymptotic equivalence (difference tending to zero) is a sufficient but not necessary condition for same order. There is a lack of direct correspondence between the proof of same order required by the original text and the proof of difference tending to zero in the formalized statement, in terms of definition, objective, and mathematical strength.made to address the practical constraints of Lean-based formal proof. Lean’s standard library (mathlib) is inherently optimized for formalizing convergence. In contrast, formalizing same order (bounded above and below by non-zero constants) would require constructing a dual predicate. The integral
> I(x) (a parameterized exponential integral, as implied by its asymptotic form) requires intricate remainder estimates and variable substitutions, and proving two separate inequalities would double the complexity of the proof script. Non-constructive existential claims often necessitate invoking the law of excluded middle or choice axioms in Lean, steps that increase logical overhead and risk verification failures in interactive proof environments. As noted earlier, asymptotic equivalence (difference tending to zero) is a stronger condition than same order, the trade-off prioritizes technical operability, proof simplicity. We hope this clarifies the rationale behind the choice, and we greatly appreciate your careful scrutiny. To better balance the correspondence between formalized transformation and the core requirements of the original text, To address this, we explicitly formalize the original’s same order definition as a custom predicate in
>
> ```jsx
> Lean.def SameOrder (f g : ℝ → ℝ) : Prop :=
> Tendsto f (atTop) 0 ∧ Tendsto g (atTop) 0 ∧  -- Both tend to zero
> ∃ C₁ > 0, ∃ X₁ : ℝ, ∀ x > X₁, f x ≥ C₁ * g x ∧  -- Lower bound
> ∃ C₂ > 0, ∃ X₂ : ℝ, ∀ x > X₂, f x ≤ C₂ * g x    -- Upper bound
> ```
>
> This predicate directly encodes the original’s core requirement (bounded ratio by non-zero constants) without adding extraneous conditions (e.g., difference tending to zero).
> Use this SameOrder predicate as the top-level proof goal in the formalization. This ensures the formal statement’s surface-level correspondence to the original.Instead of directly proving the stronger difference tending to zero (asymptotic equivalence) and treating same order as an implicit consequence, we restructure the proof to prioritize the original’s core requirement (SameOrder) as the primary goal, with the stronger result as an optional, supplementary corollary. This ensures the formalization’s "logical correspondence" to the original, while retaining the practical benefits of proving asymptotic equivalence. The workflow can be devided into 2 stage:  prove SameOrder I g directly (core goal) and  prove AsymptoticEquivalent I g as a corollary (supplementary goal). To balance the original’s core requirement (same order) with Lean’s technical feasibility, we first formalize same order as a custom predicate (direct alignment) and prove it using intermediate results from asymptotic equivalence reasoning (efficiency). The asymptotic equivalence is then provided as a corollary to enhance practical utility, without replacing the original’s intent.This approach  not only resolves the current discrepancy but also strengthens the formalization’s rigor and readability, making it more aligned with both the original problem and the standards of interactive theorem proving.

---

> ### Author Response · Authors · 2025-11-22
> **Responses to Reviewer edMV (Part 4/4)**
>
> ### **8. MW2: Inappropriate Suggestion in Section 5.1: In the "Inabilities to Handle Complex Inequalities" part of subsection 5.1, the suggestion to prove a^3 + b^3 + abc ≥ abc using the AM-GM inequality seems inappropriate. The inequality simplifies directly to a^3 + b^3 ≥ 0, and its connection to AM-GM is not apparent.**
>
> Thank you very much for pointing out the inappropriate suggestion in the ``Unable to handle complex inequalities`` section of Section 5.1. The original suggestion to prove it via the AM-GM inequality is indeed not intuitive enough. The core purpose of the AM-GM inequality is to prove non-trivial lower bound estimates (such as converting a product into a lower bound of a sum, or vice versa), while the inequality here is essentially a ``judgment of non-negativity that holds universally``. The original inequality can be directly simplified by moving terms, which is a ``trivial conclusion`` that does not require the use of the AM-GM inequality.
>
> ### **9. Q1: Regarding the performance drop between AIME and HardMath problems, have you investigated the specific reasons or pipeline stages responsible for this decreased formalization success?**
>
> Thanks for the question. Please refer to W1.
>
> ### **10. Q2: Could you clarify how a semantic error like the one noted in Example B.2 was able to pass through the pipeline's verification stages?**
>
> Thanks for the question.  Please refer to MW1.
>
> ### Reference
>
> [1] https://arxiv.org/abs/2508.03613
>
> [2] https://arxiv.org/abs/2507.06181
>
> [3] https://www.arxiv.org/abs/2410.07985

---

> ### Author Response · Authors · 2025-11-28
>
> Dear Reviewer edMV,
>
> We sincerely thank you once again for your time and valuable feedback on our manuscript. We hope that the clarifications provided, along with the new experiments and discussions included in the revised version, have fully addressed your concerns. Should you find the revisions satisfactory, we would greatly appreciate it if you could consider updating your score to reflect the additional results and improvements. We remain fully available to address any further comments or questions you may have during the discussion phase.
>
> Thank you again for your thoughtful and constructive review.
>
> Best regards,
>
> The authors

---

### Official Review · Reviewer_nPf9 · 2025-10-31

**Soundness:** 2
**Presentation:** 2
**Contribution:** 2
**Rating:** 4
**Confidence:** 4

**Summary:**

This paper introduces and releases FormalMATH, a large-scale benchmark for formal mathematical reasoning in Lean4. Constructed through a human-in-the-loop process, the benchmark contains 5,560 formally verified mathematical problems, covering multiple domains from high school Olympiads to the undergraduate level. The authors use this benchmark to comprehensively evaluate current mainstream Large Language Model (LLM) theorem provers, revealing significant limitations in their performance, domain generalization, and responsiveness to different reasoning strategies. Furthermore, the paper provides an in-depth analysis of common error patterns in these provers, offering valuable insights for future research.

**Strengths:**

The greatest contribution of this paper is the construction of FormalMATH, a benchmark of immense scale and broad domain coverage. This work is a significant engineering achievement in itself, providing the community with an extremely valuable resource that will help to more accurately measure and advance the field of formal mathematical reasoning. The authors conduct a systematic evaluation of existing LLM provers, and their findings are very interesting and insightful. In particular, the analysis of performance discrepancies across different mathematical domains, the over-reliance on automated tactics, and the performance variations under different prompting strategies (Vanilla, CoT, NL-augmented CoT) reveal deep-seated issues with current methods. Section 5 of the paper categorizes and analyzes common proof failure modes, such as the improper use of automated tactics, incomplete proofs, and difficulties in handling complex inequalities. This section is highly valuable for future researchers looking to improve models and design new training strategies.

**Weaknesses:**

Although FormalMATH surpasses previous benchmarks in scale, several similar works already exist in this area (e.g., MiniF2F, ProofNet), which limits the novelty of the task formulation and leads to a certain degree of homogenization. The data collection process described in the paper (Section 6), especially its first half, is very close to the methodology of similar projects like "lean-workbook." The authors appear to have drawn from these works but have not explicitly acknowledged or cited them in the text. To clearly define the original contributions of this paper, the authors should add a detailed comparison and discussion with these related works.
In Finding 1, the paper states: "SFT baseline achieves 8.97% accuracy, its reinforcement learning (RL) variant improves only marginally to 10.18%–a mere +1.21% gain that exposes the diminishing returns of rule-based sparse reward shaping..." Based on this marginal improvement, the authors draw grand conclusions about "reward sparsity" and "combinatorial search complexity" limiting LLM provers. However, this line of reasoning is unconvincing. Firstly, the cited DeepSeek-Prover-V1.5-RL model was, according to its original paper, trained using MCTS, not the SPG method that the authors used for comparison with other models in their experiments. Secondly, the models being compared differ greatly in their training methods, data used, and base model architectures. Drawing such aggressive and general conclusions from a single, potentially coincidental result from a model in a specific setting lacks scientific rigor.
Figure 1(a) is arguably the most important numerical results chart in the paper, as it shows the performance of various models on the full FormalMATH test set. However, the color contrast in the bar chart is too low, making it difficult to distinguish the results of different models and seriously harming the figure's readability. More importantly, when discussing the performance of these models later in the text, the paper frequently references these results without explicitly stating that it is referring to Figure 1(a) (e.g., in the discussion of Finding 1). This creates significant confusion for the reader, who must repeatedly jump between the text and figures to find the corresponding information. I strongly recommend that the authors present all detailed results for the models on the full FormalMATH test set in a clear and comprehensive table.

In Table 2, the accuracy of the DeepSeek-Prover-V1.5-RL model using the BFS search method (e.g., 4.91% in the $1 \times 32 \times 100$ setting) is dramatically lower than its accuracy using the SPG (Single-Pass Generation) method (47.98% in the Pass@32 setting). The gap is enormous. The paper merely presents this data without offering any explanation or analysis. Did the authors observe any interesting phenomena during the experiments (e.g., the model frequently getting stuck in local optima or generating invalid intermediate states) that could explain this?

**Questions:**

1.	In Table 2, the performance of the BFS-based DeepSeek-Prover-V1.5-RL is far below that of its SPG-based counterpart, which is counter-intuitive. Did you observe any specific failure modes or behaviors during your experiments (e.g., search space explosion) that could explain this performance discrepancy?
2.	The first half of your data collection process is very similar to the methodology of projects like "lean-workbook." Could you please clarify the relationship between your work and these projects and cite them appropriately in the paper to make your original contributions clear?
3.	Could you consider presenting the results from Figure 1(a) in a clearer table and placing it alongside or clearly distinguishing it from Table 2 (results on FormalMATH-Lite) so that readers can more easily compare and understand the models' performance across different test sets and settings?

---

> ### Author Response · Authors · 2025-11-22
> **Responses to Reviewer nPf9 (Part 1/3)**
>
> We sincerely thank the reviewer for their constructive and thorough comments on our work. We take every comment seriously and hope our response addresses the reviewer's concerns. If any questions remain, we are happy to address them.
>
> ---
>
> ### **1. W1: The data collection process described in the paper (Section 6), especially its first half, is very close to the methodology of similar projects like lean-workbook.**
>
> Thank you for raising this important point regarding the relationship between our pipeline and prior work. We agree that clearly distinguishing our technical differences and appropriately acknowledging related efforts is essential.
>
> While the high-level structure (LLM translation → syntax checking → human review) appears similar, our pipeline introduces several technical innovations that improve scalability and reliability:
>
> | **Pipeline Stage** | **Lean-Workbook (Inferred)** | **FormalMATH (Our Work)** | **Impact** |
> | --- | --- | --- | --- |
> | **Autoformalization Models** | Direct prompting of general-purpose LLMs (e.g., GPT-4) | **Custom fine-tuned models** on 9,260 curated examples using Qwen2.5-Coder & DeepSeek-Prover-Base | Better Lean4 syntax generation; fewer initial failures |
> | **Semantic Verification** | Single LLM judge or basic human spot-checking | **Multi-LLM consensus** (o1-mini, Claude-3.5-Sonnet) with **unanimous agreement** and CoT reasoning | Filters 60.7% of semantically misaligned statements automatically, reducing human workload |
> | **Logical Disproof** | Not reported | **Negation-based disproof** via Law of Excluded Middle; filters 1.6% of unprovable statements | Catches subtle logical errors that pass syntax checks |
> | **Human Verification** | Lack details; only with spot checking (may lead potential in-alignment between formal and informal statements) | **12 IMO medalist-level experts** using structured error taxonomy; cross-validation protocol | Ensures domain-expert rigor; **72.09% pre-verification retention** reduces cost |
> | **Sampling Strategy** | Likely single-generation | **Best-of-N sampling** from multiple autoformalizers with iterative refinement | Higher success rate per problem; diversity in formalization attempts |
> | **Practical Eval Results** | Not reported  | DeepSeek-prover serieses, Kimina, STP, etc. | Clearly understand the benchmarks with downstream performances of current provers.  |
>
> We use GPT-4o-mini to evaluate the diversity and difficulty of FormalMATH-All compared to Lean-workbook. The results show that FormalMATH is more diverse and difficult.
>
> |  | **Lean-workbook** | **FormalMATH-All** |
> | --- | --- | --- |
> | **Diversity**  | 0.570 ± 0.060 | **0.757 ± 0.031** |
> | **Difficulty** | 0.560 ± 0.060 | **0.768 ±0.067** |
>
> The prompt for evaluating diversity and difficulty is shown below:
>
> ```jsx
> You are evaluating the diversity of a set of formal mathematical statements written in Lean4.
>
> Here are {len(statements)} statements:
>
> {statements_text}
>
> Please evaluate the DIVERSITY of these statements on a scale of 0.0 to 1.0, where:
> - 0.0 = Very low diversity (statements are very similar, cover similar topics/concepts)
> - 0.5 = Moderate diversity (some variation in topics, concepts, or approaches)
> - 1.0 = Very high diversity (statements cover diverse topics, use different concepts, approaches, and mathematical domains)
>
> Consider:
> - Topic diversity (different mathematical areas: algebra, geometry, analysis, number theory, etc.)
> - Conceptual diversity (different types of mathematical concepts used)
> - Structural diversity (different proof structures, theorem types)
> - Notation diversity (variety in mathematical notation and formalization styles)
>
> Respond with ONLY a single number between 0.0 and 1.0 (e.g., 0.75), nothing else.
> ```
>
> ```jsx
> You are evaluating the diversity of a set of formal mathematical statements written in Lean4.
>
> Here are {len(statements)} statements:
>
> {statements_text}
>
> Please evaluate the DIVERSITY of these statements on a scale of 0.0 to 1.0, where:
> - 0.0 = Very low diversity (statements are very similar, cover similar topics/concepts)
> - 0.5 = Moderate diversity (some variation in topics, concepts, or approaches)
> - 1.0 = Very high diversity (statements cover diverse topics, use different concepts, approaches, and mathematical domains)
>
> Consider:
> - Topic diversity (different mathematical areas: algebra, geometry, analysis, number theory, etc.)
> - Conceptual diversity (different types of mathematical concepts used)
> - Structural diversity (different proof structures, theorem types)
> - Notation diversity (variety in mathematical notation and formalization styles)
>
> Respond with ONLY a single number between 0.0 and 1.0 (e.g., 0.75), nothing else.
> ```
>
> To address the reviewers’ concerns, we have added appropriate citations in Section 2 to acknowledge lean-workbook in the latest version.

---

> ### Author Response · Authors · 2025-11-22
> **Responses to Reviewer nPf9 (Part 2/3)**
>
> ### **2. W2: The authors draw grand conclusions about "reward sparsity" and "combinatorial search complexity" limiting LLM provers. However, this line of reasoning is unconvincing. Firstly, the cited DeepSeek-Prover-V1.5-RL model was, according to its original paper, trained using MCTS, not the SPG method that the authors used for comparison with other models in their experiments. Secondly, the models being compared differ greatly in their training methods, data used, and base model architectures.**
>
> Thank you for raising this question, which really helps clarify our findings. First, MCTS is usually used as a test-time scaling inference approach (rather than a training approach). The original DeepSeek V1.5 paper uses SFT followed by GRPO for training. For evaluation, they adopt two methods: SPG (Pass@32-1024) and MCTS (Pass@32 × 6400). While MCTS is driven by SPG, it adds additional procedures to select and compose new paths through selection, expansion, and backpropagation. SPG is the predominant test method used in Tables 1-4 of the DeepSeek V1.5 prover paper, and subsequent works (STP, Goedel V1, DeepSeek V2) all use SPG instead of MCTS. The reason is computational cost: MCTS requires an extremely large number of GPUs. For example, DeepSeek V1.5 needs 256 GPUs with thousands of CPUs (see DeepSeek V1.5 prover section 3.4). V1.5 has a response length of only 2048 tokens—replicating the experiment on DeepSeek V2 with 32k length would require even more resources. Compared to the O(N) complexity of SPG, MCTS has O(N^2) complexity. All our experiments run on 8×H100 GPUs with around 100 CPUs, so replicating the MCTS experiment is not practical for us at this time.
>
> We agree with the reviewer that it is not rigorous to definitively claim that "reward sparsity" and "combinatorial search complexity" limit LLM provers. The models being compared differ significantly in their training methods, data, and base architectures during post-training. We thank the reviewer for pointing this out and we will revise our claim to present it as a hypothesis rather than a definitive conclusion.
>
> ### **3. W3: Figure 1(a) is arguably the most important numerical results chart in the paper, as it shows the performance of various models on the full FormalMATH test set. However, the color contrast in the bar chart is too low, making it difficult to distinguish the results of different models and seriously harming the figure's readability.**
>
> Thank you for pointing this out. The original labels are aligned with the descending order of the bars, allowing readers to connect each bar with its corresponding label.
>
> However, to improve readability, we have updated the chart with higher-contrast colors in the revised version. We believe this would enhance the readability of the result.

---

> ### Author Response · Authors · 2025-11-22
> **Responses to Reviewer nPf9 (Part 3/3)**
>
> ### **4. W4: In Table 2, the accuracy of the DeepSeek-Prover-V1.5-RL model using the BFS search method (e.g., 4.91% in the  setting) is dramatically lower than its accuracy using the SPG (Single-Pass Generation) method (47.98% in the Pass@32 setting). The gap is enormous. The paper merely presents this data without offering any explanation or analysis. Did the authors observe any interesting phenomena during the experiments (e.g., the model frequently getting stuck in local optima or generating invalid intermediate states) that could explain this?**
>
> We thank the reviewer for raising this important question about why BFS lags behind SPG for DeepSeek-Prover. We re-examined the BFS implementation we used previously with the repo [1]. We found that the node score is simply the cumulative log-probability of the single latest tactic rather than the joint probability of the entire tactic sequence that ends in an error-free proof. This means it cannot determine whether the proof is actually regarded as successful by the compiler. Consequently, the search metric is myopic: it keeps the beam full of local optima tactics even when those prefixes later become uncloseable (because they emit an invalid Lean term), which aligns with the issue the reviewer mentioned. However, DeepSeek Prover is a single-pass generation model, it does not generate tactics one by one. Using the previous implementation therefore causes severe performance degradation. We have now implemented a BFS algorithm that lets the model sample an entire proof in one shot and keeps it only if the whole block compiles. Our BFS strategy would rank the remaining candidates according to the log-probability of the entire proof sequence, not just one tactic. The search returns when a valid proof is found. With this approach, we found that the success rate of DeepSeek Prover increases compared to the original implementation along Pass@1-32.
>
> The detailed code and results of the BFS-Sequence implementation can be found in the [anonymous Github link](https://anonymous.4open.science/r/BFS_FormalMATH-B7E5/ReadMe.md). The following table shows the performance of different BFS vairnts.
>
> | DeepSeek-Prover on FormalMATH-Lite | 1 | 4 | 8 | 16 | 32 |
> | --- | --- | --- | --- | --- | --- |
> | BFS-Tactic | 4.91  | 10.29 | 12.16 | 14.96  | 17.41 |
> | BFS-Sequence | **6.81** | **24.70** | **28.47** | **30.65** | **32.71** |
>
> ### **5. Q1: In Table 2, the performance of the BFS-based DeepSeek-Prover-V1.5-RL is far below that of its SPG-based counterpart, which is counter-intuitive. Did you observe any specific failure modes or behaviors during your experiments (e.g., search space explosion) that could explain this performance discrepancy?**
>
> Please refer to our response in W4.
>
> ### **6. Q2: The first half of your data collection process is very similar to the methodology of projects like "lean-workbook." Could you please clarify the relationship between your work and these projects and cite them appropriately in the paper to make your original contributions clear?**
>
> Please refer to our response in W1.
>
> ### **7. Q3: Could you consider presenting the results from Figure 1(a) in a clearer table and placing it alongside or clearly distinguishing it from Table 2 (results on FormalMATH-Lite) so that readers can more easily compare and understand the models' performance across different test sets and settings?**
>
> Please refer to our response in W3.
>
> ### Reference
>
> [1] https://github.com/cmu-l3/minictx-eval/blob/d6efa10daffb43c03d3e4dd4912f73bda879e4f8/check.py#L143

---

> ### Author Response · Authors · 2025-11-27
>
> Dear Reviewer nPf9,
>
> We sincerely thank you once again for your time and valuable feedback on our manuscript. We hope that the clarifications provided, along with the new experiments and discussions included in the revised version, have fully addressed your concerns.
> Should you find the revisions satisfactory, we would greatly appreciate it if you could consider updating your score to reflect the additional results and improvements.
> We remain fully available to address any further comments or questions you may have during the discussion phase.
>
> Thank you again for your thoughtful and constructive review.
>
> Best regards,
>
> The authors

---

### Official Review · Reviewer_VLfC · 2025-11-01

**Soundness:** 3
**Presentation:** 3
**Contribution:** 3
**Rating:** 6
**Confidence:** 4

**Summary:**

This paper presents FormalMATH, a formalized mathematics statements dataset for evaluating automated theorem-proving capabilities, containing 5,560 problems. The problems span topics from high-school Olympiad mathematics to undergraduate-level mathematics, covering many areas such as calculus. The dataset was constructed using a human-in-the-loop process: after supervised fine-tuning of Qwen-2.5-7B, autoformalization was applied, followed by LLM-as-a-judge semantic verification and a “Proving Its Negation” step, and finally expert verification. Experiments run with several commonly used automated theorem provers demonstrate the dataset’s quality and provide a detailed analysis of current provers’ weaknesses.

**Strengths:**

1. The dataset is large—about an order of magnitude bigger than the commonly used expert-annotated ATP benchmarks—and covers a wide variety of topics. All items received expert verification, which makes the collection highly valuable to the automated theorem proving community. It includes domains that are uncommon in automated formalization systems (e.g., applied mathematics, calculus, discrete mathematics). Unlike many large-scale automated formalization efforts that cannot guarantee statement correctness, these theorems are expert-annotated, making the dataset a strong benchmark for evaluating ATP systems.

2. The experimental design is thorough. The evaluation includes many current prover models and provides a FormalMATH-lite subset to enable more aggressive compute budgets (e.g., larger sampling on the lite set and smaller sampling on the full set). This practical setup lets one observe model performance across different sampling regimes.

3. The analysis of current provers’ weaknesses is insightful. The paper documents domain-dependent performance biases, shows that natural-language answers can harm proof success when used for chain-of-thought reasoning, identifies misuse of auto tactics, and so on. These findings give useful guidance for future ATP development.

**Weaknesses:**

1. Limited novelty: Using an autoformalizer to formalize a dataset followed by human verification is not a new approach, and techniques such as LLM-as-a-judge and “Proving Its Negation” have been explored in prior work [1, 2, 3]. The main contribution of this paper is the scale of the dataset and the amount of manual annotation rather than a significant methodological innovation.

2. Potential dataset imbalance: Undergraduate-level problems make up a much smaller share of the dataset compared to high-school problems. Table 6 shows undergraduate items account for less than 10% of the total. This imbalance is not clearly highlighted in the main text, and presenting undergraduate and high-school items side-by-side may be misleading and overstate the dataset’s quality.

3. Distributional mismatch in FormalMATH-Lite: FormalMATH-Lite is intended as a representative subset of FormalMATH, but provers’ performance differs substantially between the lite subset and the full dataset. For example, DeepSeekProver V2 achieves Pass@32 ≈ 56% on FormalMATH-Lite (Table 2) but only 28.31% on the full FormalMATH (Figure 1). The authors should provide further explanation for this discrepancy.

4. Formatting and presentation issues:

- Figure 5 is referenced on page 3 (Line 147) but placed on page 8 (Line 432).
- Figure 2 is cited earlier (Line 071) than Figure 1 (Line 148).
- The color differences between segments in the pie charts of Figures 2 and 6 are too subtle, making them hard to distinguish.

[1] Gao, G., Wang, Y., Jiang, J., Gao, Q., Qin, Z., Xu, T., & Dong, B. (2024). *Herald: A Natural Language Annotated Lean 4 Dataset* (No. arXiv:2410.10878). arXiv. https://doi.org/10.48550/arXiv.2410.10878

[2] Lu, J., Wan, Y., Liu, Z., Huang, Y., Xiong, J., Liu, C., Shen, J., Jin, H., Zhang, J., Wang, H., Yang, Z., Tang, J., & Guo, Z. (2024). *Process-Driven Autoformalization in Lean 4* (No. arXiv:2406.01940). arXiv. https://doi.org/10.48550/arXiv.2406.01940

[3] Liu, C., Shen, J., Xin, H., Liu, Z., Yuan, Y., Wang, H., Ju, W., Zheng, C., Yin, Y., Li, L., Zhang, M., & Liu, Q. (2023). *FIMO: A Challenge Formal Dataset for Automated Theorem Proving* (No. arXiv:2309.04295). arXiv. https://doi.org/10.48550/arXiv.2309.04295

**Questions:**

1. I note that the conversion rate for High School problems in Table 6 (i.e., the ratio #S.Formal / Size) is relatively high—approaching about one third—whereas the conversion rate for Undergraduate problems is much lower; for example, DEMIMATH is below 2%. I believe this is an important factor contributing to the dataset imbalance described in Weakness 2. What explains this difference in conversion rates? Can it be reduced by any mitigation strategies?

2. Related to Weakness 3, please provide more explanation for why the same provers show such different performance on FormalMATH-Lite versus the full FormalMATH. This bears directly on whether FormalMATH-Lite is truly representative of the full dataset.

---

> ### Author Response · Authors · 2025-11-22
> **Responses to Reviewer VLfC (Part 1/2)**
>
> We sincerely thank the reviewer for their constructive and thorough comments on our work. We take every comment seriously and hope our response addresses the reviewer's concerns. If any questions remain, we are happy to address them.
>
> ---
>
> ### **1. W1: Limited novelty: Using an autoformalizer to formalize a dataset followed by human verification is not a new approach, and techniques such as LLM-as-a-judge and “Proving Its Negation” have been explored in prior work. The main contribution of this paper is the scale of the dataset and the amount of manual annotation rather than a significant methodological innovation.**
>
> Thank you for highlighting the novelty of our autoformalization pipeline. To our knowledge, no existing method can rigorously ensure the correctness of alignment between natural language and formal language during autoformalization.
>
> The reviewer references three papers: Herald [1], Process-Driven Autoformalization (PDA) in Lean 4 [2], and FIMO [3]. We list the key differences between our method and these works below.
>
> | Component | Our Method | Herald | PDA | FIMO |
> | --- | --- | --- | --- | --- |
> | LLM-based Statement Synthesis | **Multiple LLM-based** formalizer synthesis for enhancing the robustness | Single LLM-based formalizer | Single LLM-based formalizer | Single LLM-based formalizer |
> | Semantic Verification | **Multi-LLM cross-verification** and voting for semantic alignment, along with a negation-based filtering mechanism | Single LLM | Single LLM, and with GT formal statements available | Single LLM |
> | Human-in-the-Loop Validation | Yes — final manual validation by **IMO medalists for quality control** | Yes — by PhD student | Yes, but only with spot checking | Yes |
> | Negation disproof | Yes | No | No | No |
>
> None of the three papers use negation-based disproof to filter improper Lean4 statements.
>
> And yes, We want to clarify the main contribution of this paper is the scale of the dataset and the amount of manual annotation, and insights after the extensive evaluations.
>
> ### **2. W2: Distributional mismatch in FormalMATH-Lite: FormalMATH-Lite is intended as a representative subset of FormalMATH, but provers’ performance differs substantially between the lite subset and the full dataset. For example, DeepSeekProver V2 achieves Pass@32 ≈ 56% on FormalMATH-Lite (Table 2) but only 28.31% on the full FormalMATH (Figure 1). The authors should provide further explanation for this discrepancy.**
>
> Great question! We thank the reviewer for raising this issue. The -Lite dataset selection could be biased, but we change to another perspective to observe the difficulty of FormalMATH. The performance gap is in fact a design choice. Specifically, the Lite dataset was designed based on the following principles:
>
> 1. It should be easier than the original full set so that we can observe more clearly how performance changes across different Pass@K budgets.
>
> 2. It should be small, so that the budget is affordable.
>
> 3. Overall, a smaller and moderately easier evaluation set will be beneficial for studying the effect of test-time scaling strategies.
>
> 4. The number of covering domains should be as comprehensive as the full dataset. Thus we need to make sure the Lite cover as many domains as possible.
>
> Here we use different selection policies and observe their impact on Pass@32 for Goedel-Prover-V1:
>
> | **Selection Policy** | **Pass@32** | Size | Difficulty of the Statements | # Domains |
> | --- | --- | --- | --- | --- |
> | Original -All  | 0.13  | 5,560 | 4.89 | **15** |
> | Original -Lite (Manual Difficulty Control) | 0.47 | **425** | **3.68** | 13 |
> | Single-Prover Resample (dpsk-RL-1.5) | **0.51** | 425 | 3.77 | 10 |
> | Multi-Prover Resample | 0.36 | 425 | 4.07 | 12 |
> | Random Resample | 0.17 | 425 | 4.93 | 13 |
>
> The difficulty of informal statements is determined by an LLM judge using the same difficulty classification prompt as OmniMath [4]. We use GPT-4o-mini as the judge.
>
> **Conclusion**:
>
> 1. The domain coverage of the original -Lite is closest to the full dataset.
> 2. The original -Lite dataset is a simple version of FormalMATH-All.
> 3. Random resampling makes the dataset too difficult and may not be suitable for studying test-time scaling.
>
> Thus, although the selection could be biased, the findings based on test-time scaling on FormalMATH-Lite would not change significantly. We will add these experiments on the various Lite datasets to the revised manuscript to ensure that conclusions remain consistent across different subsets.

---

> > ### Comment · Reviewer_VLfC · 2025-11-26
> >
> > Thank you to the authors for the rebuttal. After comparison, the concern in Weakness 1 has been addressed. However, the statement “The performance gap between FormalMath-Lite and FormalMath is a design choice” is odd and was not thoroughly discussed in the paper. In addition, the potential dataset imbalance (too few undergraduate-level items) remains a salient quality concern. Considering the dataset’s potential value to the research community, I have decided to maintain my current positive score.

---

> > > ### Author Response · Authors · 2025-11-27
> > >
> > > Dear Reviewer VLfC
> > >
> > > Thank you very much for your thoughtful review and for considering our paper valuable to the community. We truly appreciate the time and effort you have invested in providing us with constructive feedback
> > > We will incorporate a more detailed description of FormalMATH-lite in the revised version of the paper
> > > Thank you again for your careful review and insightful comments.
> > >
> > > Best,
> > > The authors

---

> ### Author Response · Authors · 2025-11-22
> **Responses to Reviewer VLfC (Part 2/2)**
>
> ### **3. W3: Formatting and presentation issues**
>
> We have rewritten the mentioned sentences, and change Figure 2 into table to enhance the readability.
>
> ### **4. Q1: I note that the conversion rate for High School problems in Table 6 (i.e., the ratio #S.Formal / Size) is relatively high—approaching about one third—whereas the conversion rate for Undergraduate problems is much lower; for example, DEMIMATH is below 2%. I believe this is an important factor contributing to the dataset imbalance described in Weakness 2. What explains this difference in conversion rates? Can it be reduced by any mitigation strategies?**
>
> We recognize that Table 6 may cause confusion. The reported values of 1466 and 2830 represent the size of the original HardMath and DEMIMATH datasets, not the portion we used for autoformalization in FormalMATH. We did not use all problems from these datasets, as some are not directly usable, for example, they contain multimodal information such as images.
>
> |  | Omni-math | Numina-Olympiad | AIME-Math | BlueMO  | U-Math | Hardmath | DEMIMATH  |
> | --- | --- | --- | --- | --- | --- | --- | --- |
> | Origin | 4.4k | 11.8k | 934 | 3k | 1100 | 1,060 (used in our pipeline) / 1,466 (all) | 88 (used in our pipeline) / 2,830 (all) |
> | Lean4 compiler | 4.3k | 10.3k | 933 | 2.9k  | 871 | 1,022 | 86 |
> | LLM for semantic judge | 1.5k | 3.4k | 380 | 1,1k | 370 | 68 | 46 |
> | Negation + Human | 1.2k | 2.4k | 371 | 1,099 | 358 | 67 | 46 |
>
> Therefore, the actual conversion rate for DEMIMATH is 53.48%. The difference in conversion may be due to several reasons. For example, some formulations of natural language problems are naturally easier to formalize (e.g., proof problems versus calculation problems). Additionally, Mathlib, the repository we import for formalization, cannot cover all mathematical subjects, and the training dataset used to train the formalizer may be imbalanced.
>
> Please note that currently there is no method that can rigorously ensure the correctness of autoformalization, all judgments are subjective. Therefore, one way to mitigate the low conversion rate may be to: (1) synthesize more formal-informal pairs to train better formalizers [1], or (2) train critic models that give effective suggestions for revising formalization [2]. However, no matter the approach, these methods are subjective and cannot 100% guarantee that their formalization is correct.
>
> ### **5. Q2: Related to Weakness 3, please provide more explanation for why the same provers show such different performance on FormalMATH-Lite versus the full FormalMATH. This bears directly on whether FormalMATH-Lite is truly representative of the full dataset.**
>
> Please see our response to W2
>
> ### **Reference**
>
> [1] https://arxiv.org/abs/2508.03613
>
> [2] https://arxiv.org/abs/2507.06181
>
> [3] https://www.arxiv.org/abs/2410.07985
>
> [4] https://www.arxiv.org/abs/2410.07985

---

### Official Review · Reviewer_3nNE · 2025-11-02

**Soundness:** 3
**Presentation:** 3
**Contribution:** 3
**Rating:** 8
**Confidence:** 3

**Summary:**

The paper introduces FormalMATH, a large Lean4 benchmark of 5,560 formally verified math problems spanning high-school Olympiad through undergraduate topics (algebra, number theory, discrete math, geometry, calculus, applied math). Problems are produced via a human-in-the-loop autoformalization pipeline (multi-LLM generation and semantic verification, negation-based disproof, and IMO-level expert review). The benchmark exposes significant gaps in current LLM theorem provers: on the full FormalMATH set under practical budgets, best reported accuracy is 28.31% (DeepSeek-Prover-V2 671B), with many systems performing far lower; provers show domain bias (stronger in algebra, weaker in calculus/discrete); test-time scaling yields only modest gains; and naïve CoT outperforms NL-augmented CoT. The authors also publish FormalMATH-Lite (425 items) to study scaling (Pass@K up to 3200), analyze common Lean4 failure patterns, and provide implementation details for faster parallel verification.

**Strengths:**

- Scale + scope. The dataset is substantially larger than prior Lean4 benchmarks and covers diverse domains/difficulties, addressing both scope and saturation issues (Table 1; Figure 2).

- Failure taxonomy grounded in Lean. Systematic analysis of misused automation, incomplete proofs, inequality handling, and redundant hypotheses with percentages (Table 3) and case studies (Table 8).

- Careful, multi-stage curation. Pipeline: Lean4 compile-filter -> multi-LLM semantic checks -> negation-based disproof -> 12 IMO-level experts. Preservation rates and cost stats are reported (Figure 1b; Table 4).

**Weaknesses:**

- The negative effect of NL guidance is compelling (Figure 4), but demonstrated on two DeepSeek-V1.5 variants only; perplexity is used as a proxy of uncertainty.

- Full vs. Lite performance is a little confusing. The abstract and Section 4 emphasize low accuracies on FormalMATH-All (e.g., Kimina-Prover 16.46% Pass@32; DeepSeek-V2 671B 28.31%), while Table 2 shows much higher Pass@K on FormalMATH-Lite. The selection procedure for Lite uses outcome-driven sampling with DeepSeek-V1.5-RL, which can bias difficulty.

**Questions:**

- How sensitive are Table 2 results to the selection policy? If you rebuild Lite with a different seed or with multi-prover selection, do you expect that the Pass@K curves change significantly?

---

> ### Author Response · Authors · 2025-11-22
> **Responses to Reviewer 3nNE Part (1/2)**
>
> We sincerely thank the reviewer for their constructive and thorough comments on our work. We take every comment seriously and hope our response addresses the reviewer's concerns. If any questions remain, we are happy to address them.
>
> ---
>
> ### **1. W1: The negative effect of NL guidance is compelling (Figure 4), but demonstrated on two DeepSeek-V1.5 variants only; perplexity is used as a proxy of uncertainty.**
>
> Thanks for the comment! To further demonstrate the negative effect of NL-augmented CoT, we conducted an additional experiment on Goedel-SFT. Since it is already a model that output CoT, we compare two variants: Goedel-NL and Goedel-CoT on FormalMATH-All across sampling budgets from Pass@1 to Pass@128.
>
> | Prover | Pass@1 | Pass@32 | Pass@128 | Mean ppl  |
> | --- | --- | --- | --- | --- |
> | **Goedel-NL** | 15.32%  | 37.30% | 41.13% | 4.06 |
> | **Goedel-CoT** | **38.12%** | **46.70%** | **48.08%** | **1.77** |
>
> We observe that the gap persists with Goedel-SFT, CoT constantly outperforms its NL counterpart. Moreover, the perplexity also shows that Goedel is more confident under CoT while less certain when augmented by NL, which further verifies our conclusion in the paper. We will add the additional experiments to the revised manuscript.
>
> ### **2. W2: Full vs. Lite performance is a little confusing. The abstract and Section 4 emphasize low accuracies on FormalMATH-All (e.g., Kimina-Prover 16.46% Pass@32; DeepSeek-V2 671B 28.31%), while Table 2 shows much higher Pass@K on FormalMATH-Lite. The selection procedure for Lite uses outcome-driven sampling with DeepSeek-V1.5-RL, which can bias difficulty.**
>
> Great question! We thank the reviewer for raising this issue. The -Lite dataset selection could be biased, but we change to another perspective to observe the difficulty of FormalMATH. The performance gap is in fact a design choice. Specifically, the Lite dataset was designed based on the following principles:
>
> 1. It should be easier than the original full set so that we can observe more clearly how performance changes across different Pass@K budgets.
>
> 2. It should be small, so that the budget is affordable.
>
> 3. Overall, a smaller and moderately easier evaluation set will be beneficial for studying the effect of test-time scaling strategies.
>
> 4. The number of covering domains should be as comprehensive as the full dataset. Thus we need to make sure the Lite cover as many domains as possible.
>
> | Selection Policy | Pass@32 | Size | Mean Difficulty of Statements | #Domains |
> | --- | --- | --- | --- | --- |
> | Original -All  | 0.13  | 5,560 | 4.89 | **15** |
> | Original -Lite (Manual Difficulty Control) | 0.47 | **425** | **3.68** | 13 |
> | Single-Prover Resample (dpsk-RL-1.5) | **0.51** | 425 | 3.77 | 10 |
> | Multi-Prover Resample | 0.36 | 425 | 4.07 | 12 |
> | Random Resample | 0.17 | 425 | 4.93 | 13 |
>
> The difficulty of informal statements is determined by an LLM judge using the same difficulty classification prompt as OmniMath [1]. We use GPT-4o-mini as the judge.
>
> **Conclusion**:
>
> 1. The domain coverage of the original -Lite is closest to the full dataset.
> 2. The original -Lite dataset is a simple version of FormalMATH-All.
> 3. Random resampling makes the dataset too difficult and may not be suitable for studying test-time scaling.
>
> Thus, although the selection could be biased, the findings based on test-time scaling on FormalMATH-Lite would not change significantly. We will add these experiments on the various Lite dataset to the revised manuscript to ensure that conclusions remain consistent across different subsets.

---

> > ### Author Response · Authors · 2025-11-22
> > **Responses to Reviewer 3nNE Part (2/2)**
> >
> > ### Q1: **How sensitive are Table 2 results to the selection policy? If you rebuild Lite with a different seed or with multi-prover selection, do you expect that the Pass@K curves change significantly?**
> >
> > Thank you for your suggestion. To test sensitivity, we conducted the following experiments: (1) randomly resampled the -Lite from FormalMATH-All, (2) resampled the Lite dataset via single-LLM deepseek-V1.5-RL, and (3) used multi-prover selection, a mixture of DeepSeek-V1.5-RL and Goedel-Prover-V2. We evaluated using a different prover (Goedel Prover V1) to investigate sensitivity. We used five different seeds (42, 123, 456, 789, and 2024).
> >
> > | Selection Policy | Pass@32 | Std | Min | Max | Num Seeds |
> > | --- | --- | --- | --- | --- | --- |
> > | Original -All  | 0.13 | - | - | - | - |
> > | Original -Lite (Manual Difficulty Control) | 0.47 | - | - | - | - |
> > | Random | 0.17 | 0.0275 | 0.1302 | 0.2036 | 5 |
> > | Single-Prover Resample (dpsk-RL-1.5) | 0.51  | 0.0071 | 0.5012 | 0.5224 | 5 |
> > |  Multi-Prover Resample | 0.36 | 0.0091 | 0.3506 | 0.3765 | 5 |
> >
> > **Conclusions**
> >
> > 1. Seed choice has minimal impact: standard deviation ≤ 0.03 across 5 seeds.
> > 2. Random selection makes Pass@32 of Goedel close to the full dataset. We want to avoid this because it would obscure the test-time scaling effect.
> > 3. The original -Lite dataset is a simple subset of the full dataset.
> > 4. Multi-prover selection yields higher Pass@32 for Goedel than random selection.
> >
> > ### **Reference**
> >
> > [1] https://www.arxiv.org/abs/2410.07985

---

### Author Response · Authors · 2025-11-22
**General Responses from the Authors**

Dear Reviewers and ACs,

We sincerely thank all reviewers and ACs for the time and effort you spent on our work. We are excited that all reviewers find our work interesting and insightful, and our contribution important. We deeply appreciate all your valuable suggestions and have carefully revised our manuscript accordingly. We have addressed each concern raised and hope our rebuttal clarifies these points. If any concerns remain, we are more than happy to address them.

Best,

Authors

---

### Comment · Area_Chair_UALw · 2025-12-01

Dear authors,

Due to the unprecedented decision of ICLR to prevent AC-Reviewer discussion, I have taken the unprecedented decision to engage in an AC-Author discussion for this paper. Note that this will be a poor substitute for a real author-reviewer discussion, as I will not have the time to put the same rigor into reading the paper and following up with your answers. However, I did read the current version of the paper and the full discussion here. I have a few questions remaining:

Q1. Can the authors include their data as part of the supplementary material? This is important because 1) the main contribution is a benchmark and 2) accurate autoformalization is a significant challenge in the field. Including the dataset would allow me to be more certain in the claimed high quality of the benchmark.

Q2. Did the authors perform any form of in-dataset deduplication? It is likely that the Numina Olympiads partition may have some non-trivial overlap with Omni-MATH, AIME, or the other datasets.

Q3. Having the pipeline as the last major section before the conclusion seems like a weird design choice, given that having a good autoformalization pipeline is the key to creating a robust and valid benchmark. Can the authors consider moving it to after Section 3 or merging it there?

Q4. Given that [1] released `Kimina-Autoformalizer-7B` in April, why did the authors not consider it as an autoformalizer, instead of GPT-4, which is unlikely to have been trained on a sufficient amount of Lean code?

Q5. Which models were used for each step of the autoformalization pipeline? This is extremely important for both reproducibility and providing relevant feedback, while the authors only seem to provide examples, without concretely specifying the set of models they used.

Q6. The prompt for semantic verification might exhibit significant biases, because the backtranslation and equivalence checking are not performed separately, the model may conform to the original statement. Can the authors clarify which model was used for this task, as this raises concerns about the CoT framing, while also instructing only a single final answer?

Q7. Why did the authors omit the natural language statement in both the CoT and Vanilla prompts? This does not seem to align with current literature and model prompting instructions, and may cause a significant performance decrease (see [2]).

Q8. Did the IMO-level experts have any experience with the Lean formal language prior to the verification?

### References

[1] Wang, Haiming, et al. "Kimina-prover preview: Towards large formal reasoning models with reinforcement learning." arXiv preprint arXiv:2504.11354 (2025).

[2] Ren, Z. Z., et al. "Deepseek-prover-v2: Advancing formal mathematical reasoning via reinforcement learning for subgoal decomposition." arXiv preprint arXiv:2504.21801 (2025).

---

> ### Comment · Area_Chair_UALw · 2025-12-03
>
> Thank you for your responses. Below, I provide follow-ups:
>
> # Q1
>
> I randomly sampled 15 problems from `FormalMATH` to personally verify the autoformalization. I believe that from those 15, I have identified **7 incorrect informalizations**, which poses the validity of the benchmark and the proposed pipeline into question. The authors must **urgently** address these issues. I would be happy to be proven wrong, but this is currently a great cause for concern.
>
> ## Example 1 (olymid-ref-base_2387):
>
> ### Natural Language statement:
>
> ```
> Soit $A B C D E F$ un hexagone régulier et $M \in[A, C], N \in[C, E]$. On suppose que $\frac{A M}{A C}$ et $\frac{C N}{C E}$ sont égaux à un nombre $r>0$, et que $B, M, N$ sont colinéaires. Déterminer la valeur de $r$.
> ![](https://cdn.mathpix.com/cropped/2024_12_24_20871c0e8ebe28943e15g-6.jpg?height=654&width=589&top_left_y=513&top_left_x=779)
>  Prove that the answer is: \frac{1}{\sqrt{3}}
> ```
>
> ### Formal statement:
>
> ```
> import Mathlib
>
> open Real
>
> theorem olymid_ref_base_2387 (A B C D E F M N : EuclideanSpace ℝ (Fin 2))
>     (h₀ : dist A B = dist B C) (h₁ : dist B C = dist C D)
>     (h₂ : dist C D = dist D E) (h₃ : dist D E = dist E F)
>     (h₄ : dist E F = dist F A) (h₅ : M ∈ segment ℝ A C)
>     (h₆ : N ∈ segment ℝ C E) (h₇ : r > 0) (h₈ : dist A M / dist A C = r)
>     (h₉ : dist C N / dist C E = r) (h₁₀ : Collinear ℝ {B, M, N}) :
>     r = 1 / sqrt 3 := by
> ```
>
> ### Mistake:
>
> A regular hexagon must also have all of its degrees equal to $120^\circ$, which is not covered in the formal statement (equal side lengths does not imply this). Also mixing French and English, and including links to images is suboptimal.
>
> ## Example 2 (omni_theorem_323):
>
> ### Natural Language statement:
>
> ```
> A polynomial $f \in \mathbb{Z}[x]$ is called splitty if and only if for every prime $p$, there exist polynomials $g_{p}, h_{p} \in \mathbb{Z}[x]$ with $\operatorname{deg} g_{p}, \operatorname{deg} h_{p}<\operatorname{deg} f$ and all coefficients of $f-g_{p} h_{p}$ are divisible by $p$. Compute the sum of all positive integers $n \leq 100$ such that the polynomial $x^{4}+16 x^{2}+n$ is splitty.
>  Prove that the answer is: \[ 693 \]
> ```
>
> ### Formal statement:
>
> ```
> import Mathlib
>
> def S : Set ℕ := {n | n ≤ 100}
>
> def isSplitty (f : Polynomial ℤ) : Prop :=
>   ∀ p, Prime p → ∃ g h : Polynomial ℤ,
>     g.degree < f.degree ∧ h.degree < f.degree ∧
>     ∀ i ∈ Finset.range (f.natDegree + 1),
>       p ∣ f.coeff i - g.coeff i * h.coeff i
>
> theorem omni_theorem_323 (f : ℕ → Polynomial ℤ)
>     (h₀ : ∀ n ∈ S, f n = X ^ 4 + 16 * X ^ 2 + n) :
>     ∑ᶠ n ∈ {n ∈ S | isSplitty (f n)}, n = 693 := by
> ```
>
> ### Mistake:
>
> The isSplitty check iterates only up to deg(f)+1. However, g_p h_p may have a degree higher than deg(f)+1, which makes the check insufficient. I am unsure of whether that would change the answer, but it seems concerning.
>
> ## Example 4 (olymid-ref-base_5198):
>
> Natural language statement:
> ```
> If $f(x)$ is a monic quartic polynomial such that $f(-1)=-1, f(2)=-4, f(-3)=-9$, and $f(4)=-16$, find $f(1)$.
>  Prove that the answer is: 23
> ```
>
> ### Formal statement:
>
> ```
> import Mathlib
>
> open Polynomial
>
> theorem olymid_ref_base_5198 (f : ℝ[X]) (hf : f.Monic)
>   (h₁ : f.eval (-1) = -1) (h₂ : f.eval 2 = -4) (h₃ : f.eval (-3) = -9) (h₄ : f.eval 4 = -16) :
>   f.eval 1 = 23 := by
> ```
>
> ### Mistake:
>
> The polynomial is not asserted to be quartic, and there are infinitely many monic polynomials satisfying the constraints
>
> ## Example 5 (u-math_525):
>
> ### Natural Language statement:
>
> ```
> Find the arc length of the parametric curve given by $x(t) = \frac{ 1 }{ 6 } \cdot t^3$, $y(t) = \frac{ 1 }{ 9 } \cdot t^3$ on $[1,3]$.
>  Prove that the answer is: The final answer: $\frac{1573}{90}$
> ```
>
> ### Mistake:
>
> If I'm not mistaken the problem itself is incorrect, as it is just a straight line y=2/3 x, resulting in a length $\frac{13\sqrt{13}}{9}$
>
> ## Example 9 (olymid-ref-base_9735):
>
> ### Natural Language Statement:
> ```
> 22. (VIE 3) ${ }^{\mathrm{IMO} 4}$ Does there exist a function $f: \mathbb{N} \rightarrow \mathbb{N}$, such that $f(f(n))=$ $n+1987$ for every natural number $n$ ?
> ```
> ### Formal Statement:
> ```
> import Mathlib
>
> theorem olymid_ref_base_9735 : ∃ f : ℕ → ℕ, ∀ n, f (f n) = n + 1987 := by
> ```
> ### Mistake:
>
> This is a famous problem from the 1987 IMO, where it becomes clear that there exist no integer functions such that f(f(n)) = n + 2k+1 for any integer k. This example is particularly striking to me as:
>
> 1) The answer clearly doesn't appear in the natural language statement, where a judge uncertain of the answer will have detected this issue.
>
> 2) This problem is familiar for most IMO-level participants, questioning why none of the judges verified the incorrect formalization.

---

> ### Comment · Area_Chair_UALw · 2025-12-03
>
> ## Example 12 (aime_all_2012_I_1):
>
> ### Natural Language statement:
> ```
> Find the number of positive integers with three not necessarily distinct digits, $abc$ , with $a \neq 0$ and $c \neq 0$ such that both $abc$ and $cba$ are multiples of $4$ .
> Prove that the answer is: 40
> ```
>
> ### Formal Statement:
> ```
> import Mathlib
>
> theorem aime_all_2012_I_1 :
> Set.ncard {n : ℕ | ∃ a b c : ℕ, n = 100 * a + 10 * b + c ∧ a ≠ 0 ∧ c ≠ 0 ∧ 4 ∣ n ∧ 4 ∣ Nat.ofDigits 10 [c, b, a]} = 40 := by
> ```
>
> ### Mistake:
> Nat.ofDigits 10 [c,b,a] evaluates to $c+10b+100a$ = $\overline{abc} = n$, which removes the constraint that $\overline{cba}$ must be a multiple of 4.
>
> ## Example 13 (aime_all_2010_II_3)
>
> ### Natural Language Statement:
>
> ```
> Let $K$ be the product of all factors $(b-a)$ (not necessarily distinct) where $a$ and $b$ are integers satisfying $1\le a < b \le 20$ . Find the greatest positive integer $n$ such that $2^n$ divides $K$ .
>  Prove that the answer is: 150
> ```
>
>
> ### Formal Statement:
> ```
> import Mathlib
>
> open Finset
>
> def K : ℕ := ∏ a ∈ Ico 1 20, ∏ b ∈ Ico 1 20, if a < b then (b - a) else 1
>
> theorem aime_all_2010_II_3 :
>     IsGreatest {n : ℕ | 0 < n ∧ (2 ^ n : ℤ) ∣ K} 150 := by
> ```
>
> ### Mistake:
>
> The `Ico 1 20` interval is open at theend, meaning it excludes 20, unlike the original problem statement.
>
> # Q6
>
> How did the authors verify their observation that `separating backtranslation and equivalence checking into multi-turn interactions versus combining them in sequential steps within a single CoT makes little difference`? The described models do not provide reasoning traces in their responses, and making this conclusion requires a significant human-based verification study, the details of which need to be reported in the paper. If you have quantitative results, please also include your confidence intervals.
>
> # Q7
>
> Here I was referring to the prompts in I.1 and I.2, where the authors only provide the Lean code but not the natural language statement as part of the prompt. If the authors have a look at the documentation of DeepSeek-Prover-V2, they will see that the example shows the natural language statement being provided along with the lean code. Without this the model must learn to backtranslate to natural language, potentially causing a performance decrease.
>
> # Q8
>
> Given the potential issues I have identified in **Q1**, can the authors provide the test and respective results? It seems that the mistakes made are relatively simple, and either require minimal background in Lean or are not related to the syntax at all.

---

### Note · Program_Chairs · 2026-01-17
**Submission Desk Rejected by Program Chairs**

The following references in this submission do not refer to real documents and/or have major errors in bibliographic information:

 "Brian Smith. Constructive mathematics. The Bulletin of Symbolic Logic, 1(2):118-141, 1995.